



# Physicochemical Characterization and Source Apportionment of Arctic Ice Nucleating Particles Observed in Ny-Ålesund in Autumn 2019

Guangyu Li[1], Elise K. Wilbourn[2], Zezhen Cheng[3], Jörg Wieder[1,*], Allison Fagerson[4], Jan Henneberger[1], Ghislain Motos[5], Rita Traversi[6,7], Sarah D. Brooks[4], Mauro Mazzola[7], Swarup China[3], Athanasios Nenes[5,8], Ulrike Lohmann[1], Naruki Hiranuma[2], and Zamin A. Kanji[1]

[1]Institute for Atmospheric and Climate Science, ETH Zurich, Switzerland
[2]Department of Life, Earth and Environmental Sciences, West Texas A&M University, Canyon, Texas, USA
[3]Environmental Molecular Sciences Laboratory, Pacific Northwest National Laboratory (PNNL), Richland, Washington, USA
[4]Department of Atmospheric Sciences, Texas A&M University, Texas, USA
[5]Laboratory of Atmospheric Processes and their Impacts, School of Architecture, Civil and Environmental Engineering, Ecole Polytechnique Fédérale de Lausanne (EPFL), Lausanne, Switzerland
[6]Department of Chemistry "Ugo Schiff", University of Florence, Sesto Fiorentino, 50019 Florence, Italy
[7]Institute of Polar Sciences of the National Research Council of Italy (ISP-CNR), Venice, Italy
[8]Institute of Chemical Engineering Sciences, Foundation for Research and Technology Hellas, Patras, Greece
*now at: femtoG AG, Zurich, Switzerland
**Correspondence:** Guangyu Li (guangyu.li@env.ethz.ch) and Zamin A. Kanji (zamin.kanji@env.ethz.ch)

**Abstract.**

Ice nucleating particles (INPs) initiate primary ice formation in Arctic mixed-phase clouds (MPCs), altering cloud radiative properties and modulating precipitation. For atmospheric INPs, the complexity of their spatiotemporal variations, heterogeneous sources and evolution via intricate atmospheric interactions challenge the understanding of their impact on microphysical

processes in Arctic MPCs and induce an uncertain representation in climate models. In this work, we performed a comprehensive analysis of atmospheric aerosols at the Arctic coastal site in Ny-Ålesund (Svalbard, Norway) from October to November 2019, including their ice nucleation ability, physicochemical properties and potential sources. Overall, INP concentrations ($N_{INP}$) during the observation season were approximately up to three orders of magnitude lower compared to the global average, with several samples showing degradation of $N_{INP}$ after heat treatment, implying the presence of proteinaceous INPs.

Particle fluorescence was substantially associated with INP concentrations at warmer ice nucleation temperatures, indicating that in the far-reaching Arctic, aerosols of biogenic origin throughout the snow- and ice-free season may serve as important INP sources. In addition, case studies revealed the links between elevated $N_{INP}$ to heat-lability, fluorescence, high wind speeds originating from the ocean, augmented concentration of coarse-mode particles and abundant organics. Backward trajectory analysis demonstrated a potential connection between high-latitude dust sources and high INP concentrations, while prolonged

air mass history over the ice pack was identified for most scant INP cases. The combination of the above analyses demonstrates the abundance, physicochemical properties and potential sources of INPs in the Arctic are highly variable despite its remote location.





## 1 Introduction

Arctic regions are extremely sensitive to climate change. Over the past few decades, it has been reported that the anthropogenic
warming in the Arctic is two to three times faster than the global average (Forster et al., 2021; Wendisch et al., 2019; Serreze
and Barry, 2011), a phenomenon commonly known as Arctic amplification. Satellite observations have revealed a consider-
able retreat of Arctic sea ice extent in all seasons (Stroeve et al., 2012; Serreze et al., 2007), which is identified as one of the
principal drivers of Arctic amplification given the positive surface-albedo feedback (Screen and Simmonds, 2010; Hall, 2004).
Modeling studies (Pithan and Mauritsen, 2014; Graversen and Wang, 2009; Hall, 2004) have also verified Arctic amplifica-
tion in the absence of surface-albedo feedback. Additionally, other feedbacks are also suggested as important contributors to
Arctic amplification, including atmospheric and oceanic heat transport from the mid-latitudes (Spielhagen et al., 2011), the
greenhouse effect of additional water vapor (Graversen and Wang, 2009), lapse-rate associated with the vertical structure of
warming (Bintanja et al., 2012) and cloud feedbacks (Korolev et al., 2017; Vavrus, 2004; Intrieri et al., 2002). Cloud feed-
backs are nontrivial to Arctic amplification given the ubiquity of clouds and their potential to affect the radiative balance at
both the surface and the top of the atmosphere. However, accurate quantification and prediction of cloud-induced feedbacks to
climate change remain challenging due to the rudimentary understanding of aerosol-cloud interactions and inadequate model
representations (Forster et al., 2021; Schmale et al., 2021; Murray et al., 2021), particularly in the remote Arctic.

Low-level mixed-phase clouds (MPCs), composed of a mixture of ice and supercooled liquid water, play a critical role in
the energy budget given their spatiotemporal prevalence in the Arctic (Forster et al., 2021; Korolev et al., 2017; Morrison
et al., 2012). The phase partitioning of hydrometeors within the MPCs is an essential microphysical process that intrinsically
drives the cloud feedback because more liquid water and fewer ice crystals (i.e., the trend in the warming future) are associated
with increased cloud albedo and diminished downwelling short-wave radiation, leading to a negative cloud-phase feedback
to climate change (Lohmann and Neubauer, 2018; Storelvmo, 2017). In MPCs, where the temperatures are higher than the
onset threshold of homogeneous freezing (approximately -38 °C), primary ice formation can only be triggered with the aid of
a small subset of aerosol particles termed ice nucleating particles (INPs, e.g., Kanji et al., 2017; Vali et al., 2015). Frequently,
secondary ice production increases ice crystal concentrations to several orders of magnitude higher than the INP concentration
(Korolev et al., 2020), but also cases with ice crystal number concentrations limited by the available INPs were observed in the
Arctic (Pasquier et al., 2022b). Despite the extraordinary paucity of INPs in the troposphere and that at -15 °C, approximately
1 in $10^5$ to $10^6$ aerosol particles can act as an INP (e.g., Kanji et al., 2017; Petters and Wright, 2015), their type, abundance and
variability can indirectly affect the climate by altering the microphysical and radiative properties of MPCs (e.g., DeMott et al.,
2010; Lohmann, 2002). For instance, cloud-resolving modeling studies revealed that the liquid and ice water path (Eirund
et al., 2019), atmospheric stability (Jiang et al., 2000; Harrington et al., 1999) and precipitation (Harrington and Olsson, 2001)
in the Arctic MPCs respond sensitively to INP perturbations in abundance and efficiency, and the responses were dominant over
altering the cloud condensation nuclei (CCN) concentrations (Solomon et al., 2018). In addition, the slope of INP concentration
versus ice nucleation temperature (i.e., INP efficiency) can influence the development and radiative forcing of convective clouds
(Hawker et al., 2021), and with relatively low abundance in the Arctic, enhanced Arctic amplification was simulated given





larger and fewer ice particles in MPCs (Tan and Storelvmo, 2019). Moreover, modeling studies (Hines et al., 2021; Vignon et al., 2021; Vergara-Temprado et al., 2018) produced more realistic cloud phase separations with an adjusted microphysics scheme that better represented heterogeneous nucleation processes. Therefore, further constraints on the role of INPs and robust

representations in the cloud microphysics parameterizations in climate models are of vital importance to accurately capture the cloud feedback related to Arctic amplification.

A variety of aerosols of both terrestrial and marine origin display a range of activities as INPs. Mineral dust emitted from high latitudes, e.g., the glacial outwash plains (Tobo et al., 2019) and deserts from Iceland (Sanchez-Marroquin et al., 2020), is a significant terrestrial source of INPs in the Arctic. Mineral dust particles can typically act as INPs in MPCs regime at

temperatures below approximately -15 °C (Kanji et al., 2017; Hoose and Möhler, 2012; Murray et al., 2012). In contrast, biological INPs tend to favor heterogeneous ice nucleation at relatively warmer temperatures of above approximately -15 °C (Murray et al., 2012), and their sources can stem from land, e.g., vegetation (Conen et al., 2016), sediments from runoff (Tobo et al., 2019) and thawing permafrost (Creamean et al., 2020) or from the ocean, e.g., sea spray aerosols (SSA) (Irish et al., 2017; DeMott et al., 2016; Wilson et al., 2015), phytoplankton (Ickes et al., 2020; Hartmann et al., 2020; Creamean et al.,

2019) and bacteria productivity (Šantl Temkiv et al., 2019). In addition to the INP sources originating from the vicinity of the measurement sites, the remote effect of INP emissions from mid- to low-latitudes and long-range transport cannot be neglected (Schmale et al., 2021; Vergara-Temprado et al., 2017). In deterministic INP parameterizations, the magnitude of the cloud-phase-related feedback relies on the efficiencies of INPs due to their dependency on nucleation temperatures for different INP species (Murray et al., 2021; Hawker et al., 2021).

In this study, we aim to improve our understanding of the abundance, variability, sources, physicochemical properties and impacting factors of INPs in the Arctic based on field measurement data. We introduce the campaign information, experimental setup and instrumentations in Section 2. An overview of ambient INP measurements and characterization is presented and discussed in Sections 3.1 and 3.2, respectively, and several special case studies are demonstrated in Section 3.3. Section 4 highlights the conclusions from this study and suggests potential implications for the changing climate.

## 75  2  Methods

### 2.1  Measurement location and experimental setup

The measurement campaign of ambient INP and aerosol properties was a part of the Ny-Ålesund AeroSol Cloud ExperimeNT (NASCENT) campaign (Pasquier et al., 2022a) from October to November 2019 at Ny-Ålesund (78.9 °N, 11.9 °E). Ny-Ålesund is located on the western coast of the Svalbard Archipelago (Fig. 1[a]). Ambient INP and aerosol measurements were

conducted at two locations; in an aerosol container and Gruvebadet (GVB) observatory station (Fig. 1[b]). The atmospheric container was located at the southern edge of Ny-Ålesund town and was approximately 600 m from the shore of Kongsfjorden. GVB observatory station is located about 1 km south-west from Ny-Ålesund town and is approximately 49 m a.s.l. Local sources of pollution have a minimal influence on the measurement sites given the prevailing southeasterly and southwesterly winds during the period of measurement (Pasquier et al., 2022a).





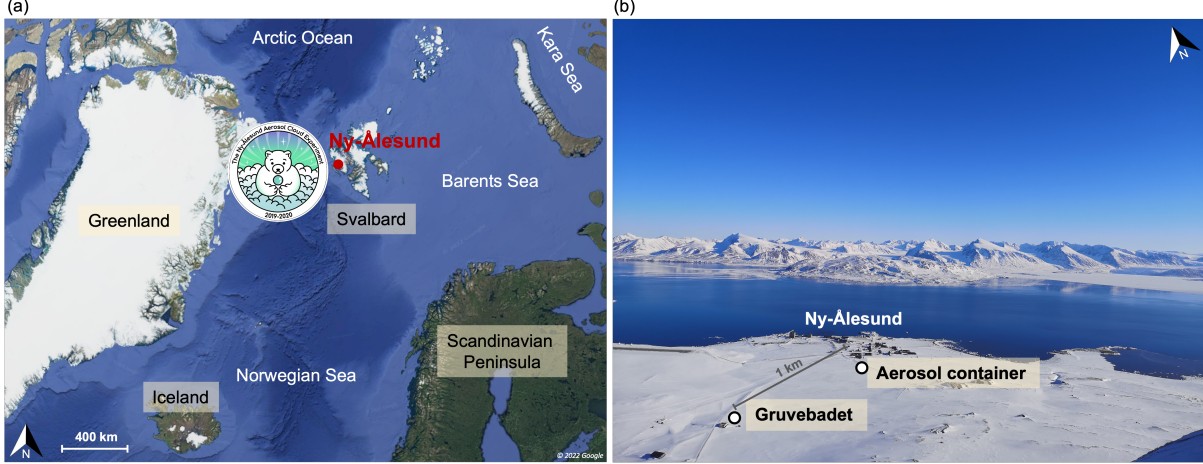

**Figure 1.** Geographic location of the sampling stations in Ny-Ålesund, Svalbard. The NASCENT campaign logo is shown in (a). The photo of (b) was taken at the mountain Zeppelin in mid-October 2019.

A flow diagram of the instrument setup is shown in Fig. 2. In the aerosol container, the aerosol flow was sampled through a total aerosol inlet mounted outside of the container, which was about 4.5 m above the ground. The inlet had an upper cut-off threshold of approximately 40 µm (Li et al., 2022) and was heated to a maximum of 40 °C to avoid clogging and frost build-up in the sampling line. The evaporation of volatile compounds in the aerosols cannot be excluded. Subsequently, the aerosol flow was directed into different branches of aerosol and INP instruments (for detailed flow configurations, see Li et al., 2022). The aerosol samples collected offline by the impinger were later subject to INP measurement via the DRoplet Ice Nuclei Counter Zurich (DRINCZ, David et al., 2019) and chemical composition analyses using computer-controlled scanning electron microscopy with energy-dispersive X-ray spectroscopy (CCSEM/EDX) and Raman microspectroscopy. At the GVB observatory, ambient INPs were analyzed using different offline techniques. Aerosol particles for analysis with DRINCZ were collected onto the 47 mm polycarbonate membrane filter (Whatman, 0.4 µm pore size) during 8-hour intervals using a low-volume aerosol sampler (LVS, DPA14, Digitel) coupled with a $PM_{10}$ inlet. The height of the inlet was approximately 5 m a.g.l., and the operating flow rate was 38.3 L min$^{-1}$. Aerosols for the West Texas Cryogenic Refrigerator Applied to Freezing Test (WT-CRAFT) analysis were collected using 47 mm polycarbonate membrane filters (Whatman, 0.2 µm pore size) during 4-day intervals (one exception was a 3-day sample started on 27 October 2019) from a central total suspended particulate (TSP) inlet with a critical-orifice-controlled sampling flow rate of 3.5 std L min$^{-1}$ (for a detailed setup see Rinaldi et al., 2021). Additionally, the aerosol properties were also monitored at the GVB observatory, including particle size distribution, black carbon and chemical compositions. The descriptions of instruments measuring INPs, aerosol physicochemical properties and meteorological conditions are given below.





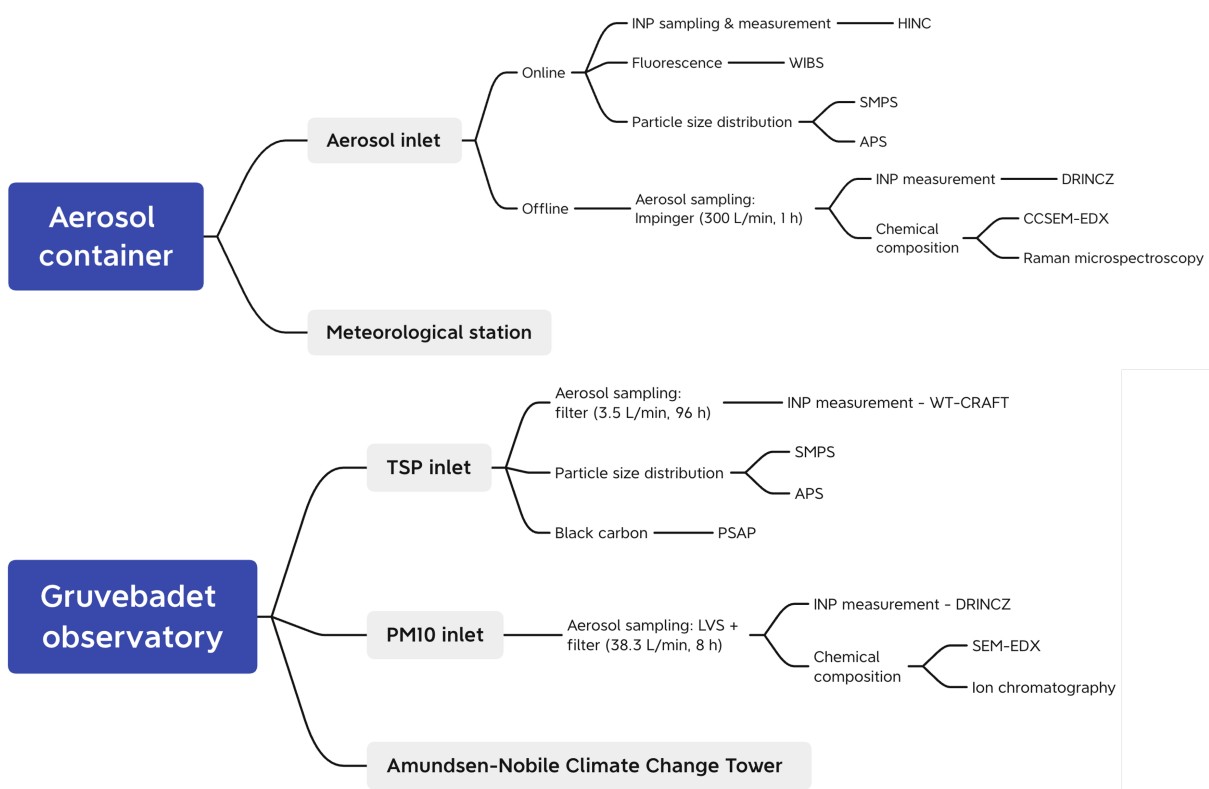

**Figure 2.** Schematic of the experimental setup in Ny-Ålesund at the aerosol container and Gruvebadet (GVB) observatory. Online means the analytical instruments with real-time aerosol sampling and monitoring, and offline denotes the instruments/devices that collect samples and take post-measurements separately. The sampling flow rates and duration are shown in the parentheses for aerosol sampling instruments. Acronyms meanings: Horizontal Ice Nucleation Chamber (HINC), Wideband Integrated Bioaerosol Sensor (WIBS), Scanning Mobility Particle Sizer (SMPS), Aerodynamic Particle Sizer (APS), DRoplet Ice Nuclei Counter Zurich (DRINCZ), Computer-Controlled Scanning Electron Microscopy with Energy-Dispersive X-ray spectroscopy (CCSEM/EDX), West Texas Cryogenic Refrigerator Applied to Freezing Test (WT-CRAFT) system, particle soot absorption photometer (PSAP) and Low Volume Sampler (LVS).

## 2.2 INP sampling and measurement techniques

### 2.2.1 DRINCZ

INP concentrations ($N_{\mathrm{INP}}$) were measured using different offline and online methods. In the aerosol container, ambient aerosols were collected into the ultrapure water (W4502-1L, Sigma-Aldrich) using the high flow-rate impinger (Coriolis® μ, Bertin Instruments, lower limit cut-off size of 0.5 μm) at a flow rate of 300 L min$^{-1}$ for 1 h. Additional ultrapure water (W4502-1L, Sigma-Aldrich) was constantly supplied to the sampling container via a refilling system during the operation of the impinger in order to compensate for the evaporation losses. The INP analysis for impinger samples was conducted onsite directly after





the sample collection. Aerosol filters collected using the $PM_{10}$ inlet at the GVB observatory were analyzed for $N_{INP}$ after the campaign in the laboratory back at ETH after frozen storage and transport at -20 °C. To determine $N_{INP}$, membrane filters were immersed in 15 mL of ultrapure water (W4502-1L, Sigma-Aldrich) and agitated using a sonicator to extract the particles into the water. During October and November 2019, a total of 137 and 77 samples were collected by the impinger and $PM_{10}$ filters respectively for immersion mode $N_{INP}$ analysis in DRINCZ (David et al., 2019). Each sample was pipetted into a sealed

Polymerase Chain Reaction (PCR) tray with 96 aliquots of 50 μL and cooled in an ethanol bath at 1 °C min$^{-1}$. During the cooling phase, a camera placed above the bath captures images of the cooling state of the PCR tray and the bath temperature was monitored. From the variation in optical brightness of an aliquot between subsequent images, the freezing temperature of the aliquots was determined. INP concentrations are derived by using the impinger flow rate and volume of an aliquot (see details in David et al., 2019). We calculated $N_{INP}$ at each integer temperature based on (Vali, 1971, 2019):

$$N_{INP}(T) = -\frac{\ln\left[1 - \frac{N_{frz}(T)}{N_{tot}}\right]}{V_a} \cdot \frac{V_{liquid}}{Q_{sample} \cdot t_{sample}} \cdot DF \tag{1}$$

where $N_{frz}(T)$ is the number of frozen aliquots at temperature $T$, $N_{tot}$ is the total number of aliquots ($N_{tot}$ = 96), $V_a$ is the volume of an individual aliquot ($V_a$ = 50 μL), $V_{liquid}$ is the volume of sampling liquid (15 mL for impinger), $Q_{sample}$ is the sampling flow rate (300 L min$^{-1}$ for impinger) and $t_{sample}$ is the sampling time (1 h for impinger). DF is the dilution factor, which was applied to quantify the dilutions for some highly IN-active samples. Diluted and non-diluted scans were

combined using the methodology provided in Wieder et al. (2022) for these samples. $N_{INP}$ of each sample derived from above were corrected for the background of blank samples based on the methods in David et al. (2019) and Li et al. (2022). For impinger samples, in order to account for contamination from the refilling system and the sampling substrate, blank samples were collected and analyzed every three days during the campaign by adding the same amount of ultrapure water for samples (15 mL) to the sampling container using the refilling system. Concerning $PM_{10}$ filter samples, empty filters were taken onsite

and reserved in the filter holders for the same duration as for the sampling stage before being stored, processed and analyzed for background INP concentrations. According to Vali (2019), all field samples were corrected for the background by subtracting the differential INP spectrum of the corresponding blanks from that of the original samples. Based on the limit of detection (LOD) of DRINCZ and purity of the nano-pure water, the highest temperature for $N_{INP}$ detection was approximately -5 °C (around which the instrument is not sensitive enough to detect the low concentrations), and the lowest temperature at which

ice nucleation could be reliably reported was -22 °C (below which $N_{INP}$ are usually closed to the background concentrations), with the overall uncertainty of the reported freezing temperature of a well of ±0.9 °C (David et al., 2019).

### 2.2.2 WT-CRAFT

The WT-CRAFT system, a replica of the Cryogenic Refrigerator Applied to Freezing Test (CRAFT) system (Tobo, 2016), which was used to estimate $N_{INP}$ in a unit volume of air for aerosol particles collected at the GVB observatory. With a de-

tection capability of > 0.003 INP std L$^{-1}$ of air, $N_{INP}$ was assessed for a total of seven samples in the temperature range of





approximately -25 °C to 0 °C, with a systematic uncertainty in freezing temperature of ±0.5 °C (Vepuri et al., 2021). The background contribution may be substantial for the WT-CRAFT $N_{INP}$ data measured below -25 °C. Alternatively, the 95 % confidence interval can represent an experimental uncertainty in the estimated $N_{INP}$ for each measured data point (Rinaldi et al., 2021). All analyses were completed within 1 year after collecting the samples, and the samples were stored in a fridge (4 °C)

before commencing the analysis.

For each experiment, the freezing properties of 70 solution droplets (3 μL each) placed on a hydrophobic Vaseline layer were assessed with a cooling rate of 1 °C min⁻¹. A cumulative number of unfrozen droplets were counted for every 0.5 °C based on the color contrast shift in the off-the-shelf video recording camera. If the freezing temperature was not obvious for any droplets, the image analysis was performed using ImageJ software to determine the temperature of phase change. Using

the same Equations. 1, $N_{INP}$ of the samples were estimated as a function of $T$, where $N_{tot}$ = 70, $V_a$ = 3 μL, and $V_{liquid}$, $Q_{sample}$ and $t_{sample}$ depend on the individual samples.

Prior to each WT-CRAFT experiment, particles on an individual filter sample were suspended in a known volume of ultrapure high-performance liquid chromatography (HPLC) grade water. The HPLC water volume was determined for the third frozen droplet to correspond to 0.003 INP L⁻¹ according to Equation. 1. It is noteworthy that we limited our WT-CRAFT data analysis

to the third frozen droplet to eliminate any uncontrollable artifacts in our WT-CRAFT data (Hiranuma et al., 2019). Because of the negligible background freezing contribution of the field blank filter at -25 °C (i.e., < 3 %), we did not apply any background corrections to our $N_{INP}$ data. Otherwise, we followed the exact same protocols described in Rinaldi et al. (2021) for our suspension generation and dilution.

### 2.2.3   HINC

To complement the INP measurements at colder temperatures, we sampled and measured $N_{INP}$ with HINC (Lacher et al., 2017), a continuous flow diffusion chamber. HINC was operated at $T$ = -30 °C (± 0.4 °C) and relative humidity with respect to water $RH_w$ = 104 % (± 1.5 %), representative for ice nucleation in immersion and condensation modes. The detailed experimental configuration of HINC can be found in Fig. 2 of Li et al. (2022). With a size threshold of 5 μm derived from the water droplet survival test (Lacher et al., 2017) at the designed experimental conditions, we were able to distinguish the ice crystals from

water droplets during the sampling phase. To account for frost particles that can be misidentified as INPs when detaching from the inner surface, we applied a routine of filtered air measurements (5 min) before and after each sampling interval (15 min) to determine the background count of ice particles and the limit of detection based on Poisson statistics. Subsequently, $N_{INP}$ was calculated by subtracting ice particle concentrations during the background interval from that during the sampling interval (see detailed calculations in Lacher et al., 2017). During the field campaign between October and November 2019, we reported

135 INP concentrations from HINC measurements that were higher than the limit of detection of the instrument out of a total of 348 observations. In other words, the 135 INP concentration data points have a significance level of 68.3 % and were more reliable for extrapolation due to the limitation of the instrument at the measurement conditions.

Using a combination of the above-mentioned INP instrumentations can provide broader coverage of particle sizes, types, as well as freezing temperatures.





## 2.3    Heat treatments

Macromolecules originating from biological species (e.g., bacteria and phytoplankton) that typically comprise of proteins can effectively catalyze ice nucleation (Pummer et al., 2015; Hill et al., 2016). Proteins are susceptible to heat, i.e., heating effectively unfolds the proteinaceous structure, degrading the IN ability of the particles (Creamean et al., 2021; Hill et al., 2016; McCluskey et al., 2018; Pummer et al., 2015). For the heat treatment, liquid samples from the impinger and washout of $PM_{10}$ filters and TSP filters (for WT-CRAFT analysis) were subjected to 95 °C for 20 min. Subsequently, after being stabilized to room temperature, they were redistributed to PCR trays for INP analysis using DRINCZ. By comparing the IN activity after heating, it is possible to assess the contribution of heat-labile species to the INP population, which could be used as a proxy to indicate the presence of biological INPs. Post-campaign heat tests were conducted in the laboratory. To elucidate the relative change of INPs affected by degradation due to freezing storage only, we repeated the INP concentration analysis for original impinger samples that were selected for heat treatment. Heat treatment was applied to all $PM_{10}$ filter samples, and 14 impinger samples overlapping with the WT-CRAFT time window for repeated INP analysis and heat treatment for comparison.

### 2.4    Particle chemical composition analysis

#### 2.4.1    CCSEM/EDX for impinger droplet residual samples

Computer-controlled scanning electron microscopy with energy-dispersive X-ray spectroscopy (CCSEM/EDX) was utilized to automatically probe the morphology and elemental composition of individual particles in a series of selected impinger samples collected on the Aluminum foils (Laskin et al., 2006). The system includes an environmental scanning electron microscope (ESEM, Quanta 3D, Thermo Fisher) equipped with an FEI Quanta digital field emission gun operated at 20 kV and 480 pA with 30 μm aperture and spot size of 6.0 nm to retrieve the ESEM images, which were used to retrieve the morphologies of individual particles (Lata et al., 2021). These individual particles are recognized based on the difference of brightness and contrast between particles and substrate in ESEM images. The ESEM was also equipped with an EDX spectrometer (EDAX, Inc.) to determine the relative percentages of 12 elements (C, N, O, Na, Mg, Si, P, S, Ca, Mn, Fe, and Zn) in the individual particles (see Table C1). Applying a k-means clustering algorithm on all analyzed particles using their atomic percentages (Hartigan and Wong, 1979), we categorized components inside each particle as Salt (Na + Mg ≥ 15 %), Si-dust (Si + Ca + Fe ≥ 15 % & Si ≥ Fe), Fe dust (Si + Ca + Fe ≥ 15 % & Fe > Si), Sulfate (S) (S ≥ 2 %), Phosphorus (P ≥ 1 %) and metal containing particles (Mn + Zn ≥ 15 %). The number of clusters was determined using the silhouette method (Kodinariya and Makwana, 2013). Then based its composition, we classified individual particle as salt containing particle, dust containing particle, metal containing particle, and P containing particle. It should be noticed that particle can be clustered into multiple classes. For instance, if a particle only fulfils Na + Mg ≥ 15 % and Si + Ca + Fe ≥ 15 % & Si ≥ Fe, its composition is classified as (salt + Si - dust). It should be noted that the chamber was operated at 293 K under vacuum conditions (ca. 2 × $10^{-6}$ Torr). Thus, volatile and semi-volatile components might have been evaporated. For selected impinger samples as case studies indicated in Section 3.3.2, the total number of particles analyzed by CCSEM-EDX were 1171, 1286 and 1016 for Impinger_high, Impinger_moderate and Impinger_low samples, respectively.



### 2.4.2 Single particulate matter chemical composition using Raman microspectroscopy on impinger samples

Selected samples were characterized with Raman microspectroscopy using a Thermo Scientific DXR Raman Spectrometer
coupled to an Olympus BX 20 microscope and a CCD to capture images of the particle as the analysis was performed (Deng et al., 2014). The analysis was performed with a 532 nm frequency doubled Neodymium-doped Yttrium orthovanadate (Nd: YVO4) diode-pumped solid-state laser with 3 Mw power. A 50× objective (Thermo Scientific) was used to find a single particle that was then sampled for bond composition following the procedures used in previous studies (Deng et al., 2014). Ten exposures at ten seconds each were averaged to smooth the sample spectrum. Samples that showed signs of fluorescence were
not analyzed past identification of the fluorescence, as there was no way to tell whether additional peaks were obscured by the fluorescence signal.

Peaks were classified based on Larkin (2017) and samples were classified into broad categories. As the environmental samples contain both internally and externally mixed aerosol particles, a more detailed classification was not possible. The samples were classified as metal-containing, nitrogen-containing, sulfur-containing, organic-containing, and aromatic-ring-
containing (note that particles can be placed into more than one category based on composition).

### 2.4.3 JEOL SEM-EDX for $PM_{10}$ filter samples

The JEOL scanning electron microscopy – energy-dispersive X-ray spectroscopy (SEM-EDX) system (Model JSM-6010LA) was used to assess the elemental composition of aerosol particles collected on the $PM_{10}$ filters. Briefly, this system allowed us to characterize the atomic percentage of 14 elements, including N, O, Na, Mg, Al, Si, P, S, Cl, K, Ca, Mn, Fe, and Zn, on a
single particle basis (see Table C2). All analyses were performed under a constant measurement condition, which is a 20 keV electron beam accelerating voltage and a 10 mm distance between the SEM objective lens and the specimen surface. Because the particles were collected on polycarbonate filters, it was not possible to determine the atomic percentage of carbon. Instead, SEM-EDX data were mainly used to determine the presence or absence of mineral dust- and/or sea salt-relevant elements using the simple particle-type classification method, which was previously applied for the Alaskan Arctic aerosol characterization
study Hiranuma et al. (2013).

A total of 627 aerosol particles (i.e., 6 filter samples and approximately 100 particles per sample) were analyzed in this study. Individual particles were assessed for their x-axis and y-axis segment diameters, and a cross-sectional average diameter was computed for each particle. The largest particle analyzed was 6.04 μm in diameter. It should be noted that the edge of filter pores can be misidentified as particles under CCSEM-EDX due to similar brightness and contrast as particles. The lower
detection limit for the JOEL SEM method is approximately 0.5 μm particle diameter. Thus, we decided to manually analyze a subset of particles with a regular SEM-EDX system. We note that the manual operation of SEM-EDX is a time-consuming and labor-intensive process, so its application during this study was limited. For this reason, a few samples were selected to study in greater detail. A subset of single particles was selected on each filter to analyze particle composition, with at least 100 randomly selected particles (at least 25 particles per 128 μm × 96 μm cross-section) across each filter. to give an approximation
of population chemical composition and major particle groups (i.e., mineral dust- or sea salt-rich particles). No specific particle



size or shape was pre-selected for analysis. Instead, a range of sizes and shapes was targeted to give the best approximation of overall population chemistry.

## 2.5 Complementary measurements and analyses

### 2.5.1 Particle size distribution

The size distribution of submicron particles was measured using a scanning mobility particle sizer (SMPS, Model 3938, comprising a 3082 classifier, a 3081 long differential mobility analyzer and a 3787 CPC, TSI Inc.). The sampling flow rate of the SMPS was 0.6 L min$^{-1}$ with a sheath-to-sample ratio of 10:1, leading to a range from approximately 15 to 600 nm in electrical mobility diameter. In addition, multiple charge correction was applied to account for the misclassification of large particles with multiple charges. Parallel to the SMPS, the size distributions of coarse-mode (ranging from approximately 0.5 to

20 μm in aerodynamic diameter) particles were analyzed by an aerodynamic particle sizer (APS, model 3321, TSI Inc.) at a flow rate of 1 L min$^{-1}$. For the aim of cross-comparison, the electrical mobility diameters obtained from SMPS and aerodynamic diameters from the APS were converted to volume-equivalent diameters assuming an average particle density of 2 g cm$^{-3}$ (Li et al., 2022; Tobo et al., 2019) and a dynamic shape factor of 1.2 (Li et al., 2022; Thomas and Charvet, 2017). A set of identical SMPS and APS was available at both GVB observatory and aerosol container for this study.

### 2.5.2 Particle fluorescence

The concentration of biological fluorescent particles with diameters ranging between 0.5 and 20 μm was monitored using the wideband integrated bioaerosol sensor (WIBS-5/NEO, DMT) on a single particle basis, downstream of the inlet of the aerosol container. The WIBS uses ultraviolet light to trigger the excitation of particles and to detect the emissions scattered from the fluorescent particles (Toprak and Schnaiter, 2013). The wavelengths of excitation and emission were specifically designed to probe biological fluorophores, e.g., trytophan-containing proteins, NAD(P)H and riboflavin, which are ubiquitous

in the airborne microbes (Pöhlker et al., 2012). The resulting total fluorescence was then measured in three fluorescence channels: FL1 (310–400 nm) and FL2 (420–650 nm) following a 280 nm excitation, and a 370 nm excitation for channel FL3 (420–650 nm). Each individual particle is identified to be fluorescent in any channel when the intensity of its fluorescence emission is higher than the baseline threshold. However, the fluorescent signals from WIBS may misidentify non-biological particles with fluorescent signatures, such as some dust particles, HULIS and PAHs (Toprak and Schnaiter, 2013). These

misclassifications could be suppressed by characterizing the fluorescence in different channels independently, which allows particles to be classified into different fluorescence categories (Savage et al., 2017; Perring et al., 2015). The new fluorescence categories are named A, B, C, AB, AC, BC and ABC based on the detection of activation in the original channels (i.e., FL1, FL2 and FL3), signifying particle fluorescence detected in channel FL1 only, FL2 only, FL3 only, FL1 and FL2, FL1 and FL3, FL2 and FL3 and all three channels, respectively. More details are given in Fig. 1 of Savage et al. (2017); Perring et al.

(2015). To minimize the false-positive signal of non-biological particles that are fluorescent, the category "AC + ABC" (a particle fluoresces in both the FL1 and the FL3 channels but is not activated in the FL2 channel) was applied as a proxy for



detecting biological fluorescent aerosol particles, with reduced remaining interference from non-biological sources (Toprak and Schnaiter, 2013).

### 2.5.3   Black carbon

Equivalent black carbon ($e$BC) concentration data were used to assess the potential contribution of BC on INP abundance. The $e$BC was measured using a particle soot absorption photometer (PSAP, Radiance Research), with which the light absorption coefficient ($b_{abs}$, m$^{-1}$) can be determined at three absorption wavelengths (Gilardoni et al., 2019). eBC concentration is derived from the light absorption coefficient at 660 nm. The mass concentration of BC ($M_{BC}$, g m$^{-3}$) can be estimated by dividing $b_{abs}$ by the constant mass absorption cross-section of BC (MAC, m$^2$ g$^{-1}$) of $10.0 \pm 0.2$ m$^2$ g$^{-1}$ (Sinha et al., 2017).

### 2.5.4   Meteorological conditions

We investigated the relationships between INP concentrations and the meteorological variables, including ambient temperature ($T_{env}$), relative humidity ($RH_{env}$), pressure ($p_{env}$), wind speed ($ws$) and wind direction ($wd$). The meteorological measurements for correlating samples from the aerosol container were conducted using an automatic meteorological station (MetSystems model WS-501, OTT). For measurements at the GVB observatory, we used the meteorological data from the Amundsen-Nobile Climate Change Tower configured with a set of meteorological sensors (details described in Mazzola et al., 2016).

### 2.5.5   Ion chromatography

The ionic compositions of the filters collected at GVB observatory in parallel with LVS PM$_{10}$ filters were measured in aqueous extracts prepared prior to analysis using the ion chromatography following the procedures described in Becagli et al. (2011). The resulting ionic compositions involve many cations, inorganic anions (see detailed ion species in Becagli et al., 2011), methanesulfonic acid (MSA) and oxalate.

### 2.5.6   Backward trajectory analysis

Air mass backward trajectories were computed with the Hybrid Single-Particle Lagrangian Integrated Trajectory (HYSPLIT) model available *online* (Rolph et al., 2017; Stein et al., 2015). Ten-day (240-h) backward trajectories were computed 5 m a.g.l from the sampling location every six hours during the sampling period. To account for wet deposition, complete particle loss was assumed at 7 mm rainfall along the trajectory (Gong et al., 2020). The rainfall was summed along the trajectory and the trajectory origin is reported as either the time and location where cumulative rainfall exceeded 7 mm or the location of the air mass 240 hours before the sampling time, whichever was earlier. Air mass origin is reported in broad geographic terms, with oceans named according to commonly accepted names as defined by the U.S. Board on Geographic Names and terrestrial regions defined by continent. The Arctic regions were defined as occurring above 60 °N latitudes. Back trajectory origins were determined, accounting for particle deposition due to wet deposition.




# 3 Results and discussion

## 3.1 Overview of atmospheric INP concentration

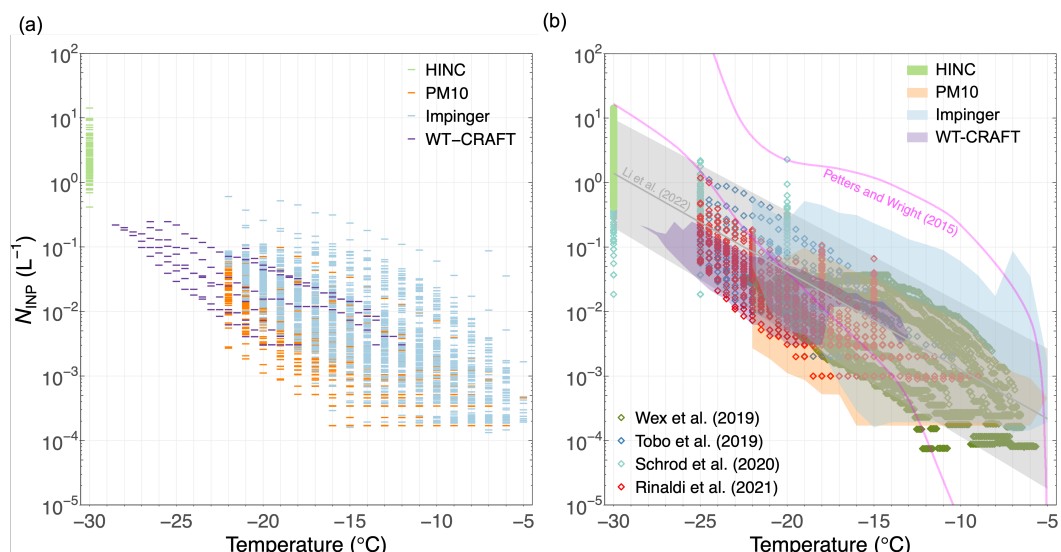

**Figure 3.** Ambient INP concentration as a function of activation temperature for (a) observations of $N_{INP}$ (symbolized with "-") including the sampling and measurements with HINC ($T$ = -30 °C) and the impinger (-22 °C $\sim$ -5 °C) at the aerosol container, and with the PM$_{10}$ filter (-22 °C $\sim$ -5 °C) and WT-CRAFT (-30 °C $\sim$ -12 °C) at GVB observatory during autumn 2019 in Ny-Ålesund. (b) comparison of our observations to literature data. The area between two lines in magenta is a compilation of INP concentrations determined from precipitation samples from the mid-latitude (Petters and Wright, 2015). The light gray line (median) and the shaded area (95 % confidence interval) denote the INP parameterization developed from the Ny-Ålesund data during the NASCENT campaign in autumn 2019 and spring 2020 (Li et al., 2022). INP concentration data from Rinaldi et al., 2021 (spring and summer), Schrod et al., 2020 (year-long), Tobo et al., 2019 (spring and summer) and Wex et al., 2019 (spring, summer and autumn) measured at the same location are also presented in colored diamonds.

Figure 3 (a) shows the overall $N_{INP}$ range as a function of ice nucleation temperature at Ny-Ålesund during the NASCENT campaign (Pasquier et al., 2022a) from October to November 2019. In general, measurements among different techniques agree with each other as demonstrated by the substantial overlap. Specifically, the median $N_{INP}$ at $T$ = -20 °C measured with the PM$_{10}$, impinger and WT-CRAFT instruments were approximately $1.0 \times 10^{-2}$, $2.7 \times 10^{-2}$ and $0.7 \times 10^{-2}$ L$^{-1}$, respectively. In addition, the variability of $N_{INP}$ as a function of $T$ changes with $T$. In particular, for the PM$_{10}$ and impinger samples, $N_{INP}$ varied over 3 orders of magnitude at mid-range activation temperatures ($T$ = -16 $\sim$ -14 °C), but the variation range shrunk as $T$ increased or decreased. This could be simply attributed to the reduced number of available observations as the lower and upper instrument detection limits are approached (see Section 2.2.1).

Compared to the global range of $N_{INP}$ derived from precipitation samples collected in the continental mid-latitudes (Petters and Wright, 2015) (3 [b]), the average $N_{INP}$ observed in the Arctic is approximately 2 orders of magnitude lower. However,



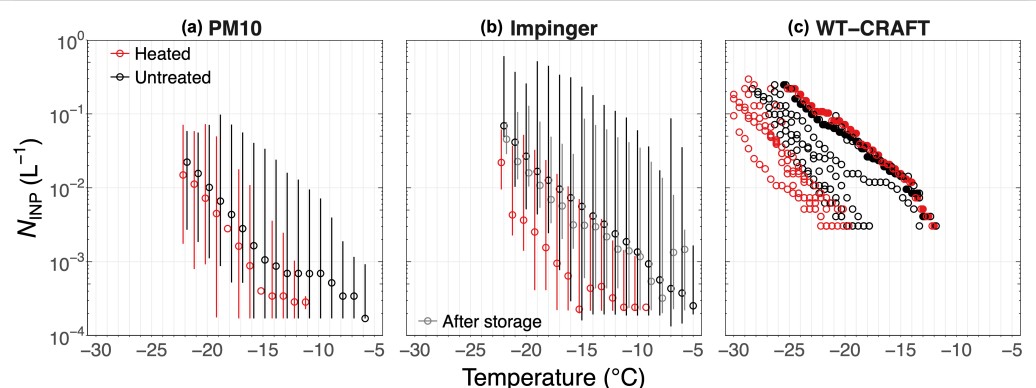

**Figure 4.** Ambient INP concentration with and without heat treatment conditions as a function of ice activation temperature for different samples: (a) PM$_{10}$, (b) impinger and (c) WT_CRAFT). The hollow circles shown in (a) and (b) represent the median $N_{INP}$ of all samples collected during October-November 2019, and vertical extensions represent the 5 - 95 % percentile of the measurements. For impinger samples shown in (b), the heat treatment was not conducted onsite. Instead, the frozen samples were reanalyzed (gray open circles) after storage in the laboratory and subjected to heat treatment. The open circles shown in (c) denote the 7 individual INP spectra during the autumn 2019 campaign in Ny-Ålesund (from 12: 43 UTC 03/10/2019 to 12:49 UTC 30/10/2019 at an approximately 4-day sampling interval). The filled circles in (c) represent the sample taken from 10:37 UTC 19/10/2019 to 12:15 UTC 23/10/2019 for original INP analysis and reanalysis after heating.

at the highest $T$ (-5 to -7 °C), $N_{INP}$ observed with the impinger were close to or higher than the global average level (Petters
and Wright, 2015), despite high uncertainties may arise. This indicates a potentially significant contribution of biological INPs in the Arctic. The Arctic INP parameterization (Li et al., 2022) derived from the same campaign developed from impinger and HINC data from both autumn 2019 and spring 2020 generally agreed with the observations from PM$_{10}$ and WT-CRAFT. Moreover, Figure 3 (b) also displays a compilation of $N_{INP}$ measurements from recent ground-based observations in Ny-Ålesund for comparison. Note that besides the natural variability of $N_{INP}$, a number of factors (e.g., systematic error of each
instrument, sampling volume and seasonality) could contribute to the differences in observed $N_{INP}$ among studies. Nevertheless, the majority of literature $N_{INP}$ overlapped with our observation range, except Schrod et al. (2020) who observed systematically higher $N_{INP}$ at $T$ = -20 °C, and data from Wex et al. (2019) at higher $T$ who measured lower INP concentrations, likely due to different sensitivities of the instruments and the seasons during which they took measurements because a different season could imply different sources and abundance of INPs.
To detect the presence of proteinaceous biological ice-nucleating entities, heat treatment was applied to sample solutions. Figure 4 compares the overall INP spectra between different untreated and heated samples using different sampling and INP instruments. The range of activation temperatures measured is different for WT-CRAFT and PM$_{10}$ and impinger samples, likely due to the different instrumentation limits (i.e., droplet volume). For untreated samples, the overall INP concentrations at the same activation temperatures sampled by impinger from the aerosol container were systematically higher (ca. 2 to 3
times) compared to those sampled by PM$_{10}$ filters at the GVB observatory. At the GVB observatory, $N_{INP}$ sampled with the





TSP inlet for WT-CRAFT analysis were comparable to those measured from $PM_{10}$ filters in the overlapping temperature range. Despite the longer sampling duration (> 3 days), this study includes the WT-CRAFT $N_{INP}$ data as it compensates the immersion freezing data lower than the temperature range measured by DRINCZ and bridges the gap towards HINC measurements. The differences in $N_{INP}$ between the impinger samples which were analyzed for $N_{INP}$ directly after the collection on site and the

$PM_{10}$ filters which we believe is due to the frozen storage at -20 °C and analyzed for $N_{INP}$ 1.5 years after the campaign in the laboratory. Beall et al. (2020) revealed that heat-labile INPs tend to be more sensitive to storage, i.e., storage and re-thawing of heat-sensitive samples in the case for our $PM_{10}$ samples (Figure 4 [a]), which could conceivably degrade the IN activity (although the extent is unknown). This hypothesis also held true for impinger samples (see Fig. 4 [b]) when reanalyzed in the laboratory ("after storage" in gray). Here overall $N_{INP}$ decreased nearly at all investigated temperatures. Additionally, with

different sampling flow rates and time, the total volume of air collected by each impinger, $PM_{10}$ and WT-CRAFT filter sample was approximately 18 m$^3$, 18.4 m$^3$ and 20.2 m$^3$, respectively. Given the similar volume of air collected in each sample, another possible reason for the differences in INP population was due to different size thresholds based on the sampling efficiency (i.e., impinger: > 0.5 μm, $PM_{10}$ filters: < 10 μm, TSP inlet for WT-CRAFT: > 0.2 μm) at which the aerosols were collected using the three sampling techniques. In particular, aerosol samples collected by the impinger included particles larger than 10 μm, which

are frequently effective INPs given their large sizes (DeMott et al., 2015; Mason et al., 2016) despite their relative scarcity in the ambient air. In terms of heat sensitivity, the general degradation of $N_{INP}$ for all samples after being heated at 95 °C for 20 minutes, particularly at $T \geq$ -15 °C, suggests that heat-labile IN active proteins or biological macromolecules likely contribute to INP sources in the Arctic during our measurement season. The remaining IN active materials could be mineral dust or heat-resistant organics (Conen et al., 2016; Hill et al., 2016), for which $N_{INP}$ were similar for $PM_{10}$ and impinger samples.

Specifically, one interesting exception highlighted in Fig. 4 (c) (filled circles) displayed both high $N_{INP}$ and heat-insensitivity for the sample collected from 19/10 to 23/10 in 2019. More evidence will be provided by measurements with finer resolutions to unveil the INP abundance and potential sources (see Section 3.3).

### 3.2 $N_{INP}$ correlations with meteorological and aerosol physicochemical variables

Table 1 summarizes the Pearson correlation coefficients ($r$) between the entire time series of $N_{INP}$ and several physicochemical

and meteorological variables. It can be seen that for most cases, $-0.5 < \rho < 0.5$, suggesting no strong correlation in general. The absence of a strong correlation demonstrates that during the 6-week measurement campaign, a mixture of parameters correlated with the observed INP population. However, the actual level of correlations could be underestimated for all cases, because in order to synchronize $N_{INP}$ and other measured parameters with different time resolutions, the variability could be concealed when averaging the variable over a period of time, e.g., averaging 3-min particle size distribution data for comparison

with $N_{INP}$ over several hours in a high-volume filter sample. In addition, the level of statistical significance was reduced due to the shrunk data sets. The aforementioned rationale supports the need for high time resolution of $N_{INP}$ measurements.



**Table 1.** Pearson correlation coefficients ($r$) between INP concentration at different nucleation temperatures (sampled by impinger [-6 ∼ -18 °C] and HINC [-30 °C] at the aerosol container, separated by the double solid lines in the upper part of the table) to aerosol physicochemical and meteorological variables: aerosol concentration of particles within different diameter ranges ($n_{0.01∼0.1}$, $n_{0.1∼0.5}$, $n_{>0.5}$, $n_{0.5∼1}$, $n_{1∼2.5}$ and $n_{>2.5}$, with diameters in μm), aerosol surface area concentration of particles with diameter at different size ranges ($S_{>0.5}$ and $S_{<2.5}$, with size unit in μm), equivalent black carbon concentration ($e$BC), fluorescent particle concentration defined by different categories ($n_{fluor}$, $n_{FL1}$, $n_{FL2}$, $n_{FL3}$ and $n_{AC+ABC}$, according to the criteria of classification defined in Section 2.5.2) and meteorological variables (ambient temperature [$T_{env}$], RH [$RH_{env}$], pressure [$p_{env}$], wind speed [$ws$] and direction [$wd$]). Note that different aerosol size-resolved variables were correlated with $N_{INP}$ at different nucleation temperatures due to different size cut-offs of INP sampling/measurement, impinger sampled $N_{INP}$ at $T$ = -6, -9, -12, -15, -18 °C with a lower size threshold of 0.5 μm, and HINC measured $N_{INP}$ at $T$ = -30 °C with an upper size limit of 2.5 μm, located in the aerosol container. Correlations between $N_{INP}$ (-6 ∼ -18 °C) and sodium, ammonium, calcium, nitrate, sulfate and methanesulfonic acid (MSA) were derived from the filter samples collected at GVB in parallel with PM$_{10}$ INP filters using the ion chromatography analyses (separated by the double solid lines in the lower part of the table).

| Variable | $N_{INP}$ (T=-6°C) | $N_{INP}$ (T=-9°C) | $N_{INP}$ (T=-12°C) | $N_{INP}$ (T=-15°C) | $N_{INP}$ (T=-18°C) | $N_{INP}$ (T=-30°C) |
|---|---|---|---|---|---|---|
| $n_{0.01∼0.1}$ (L$^{-1}$) | - | - | - | - | - | 0.10 |
| $n_{0.1∼0.5}$ (L$^{-1}$) | - | - | - | - | - | 0.01 |
| $n_{>0.5}$ (L$^{-1}$) | 0.24 | **0.23** | 0.14 | 0.15 | 0.09 | - |
| $n_{0.5∼1}$ (L$^{-1}$) | 0.24 | **0.21** | 0.13 | 0.13 | 0.07 | 0.17 |
| $n_{1∼2.5}$ (L$^{-1}$) | 0.20 | **0.24** | 0.15 | **0.17** | 0.10 | **0.18** |
| $n_{>2.5}$ (L$^{-1}$) | 0.26 | **0.31*** | **0.27** | **0.28** | 0.22 | - |
| $n_{tot}$ (L$^{-1}$) | 0.08 | -0.04 | 0.00 | -0.01 | 0.11 | 0.06 |
| $S_{>0.5}$ (m$^2$ L$^{-1}$) | 0.25 | **0.28** | **0.20** | **0.22** | 0.14 | - |
| $S_{<2.5}$ (m$^2$ L$^{-1}$) | - | - | - | - | - | 0.13 |
| $e$BC (ng m$^{-3}$) | **0.09** | -0.07 | -0.04 | -0.05 | -0.03 | 0.12 |
| $n_{fluor}$ (L$^{-1}$) | **0.55**** | **0.36*** | **0.34*** | **0.38*** | **0.35*** | **0.20** |
| $n_{FL1}$ (L$^{-1}$) | **0.55**** | **0.23** | **0.28** | **0.31*** | **0.28** | **0.18** |
| $n_{FL2}$ (L$^{-1}$) | **0.56**** | **0.29** | **0.27** | **0.29** | **0.27** | 0.17 |
| $n_{FL3}$ (L$^{-1}$) | **0.48*** | **0.40*** | **0.36*** | **0.39*** | **0.37*** | 0.17 |
| $n_{AC+ABC}$ (L$^{-1}$) | **0.63** | **0.29** | **0.32*** | **0.31*** | **0.20** | 0.14 |
| $T_{env}$ (°C) | -0.19 | -0.02 | 0.00 | 0.06 | 0.03 | **0.18** |
| $RH_{env}$ (%) | -0.12 | 0.19 | 0.16 | **0.21** | 0.10 | 0.12 |
| $p_{env}$ (hPa) | 0.00 | -0.13 | -0.19 | **-0.24** | **-0.29** | -0.11 |
| $ws$ (m s$^{-1}$) | **0.47*** | **0.52**** | **0.49*** | **0.52**** | **0.42*** | 0.17 |
| $wd$ (°) | **0.30*** | **0.33*** | **0.24** | **0.20** | **0.22** | 0.07 |
| Sodium (ng m$^{-3}$) | N.A. | -0.12 | -0.15 | 0.00 | 0.04 | - |
| Ammonium (ng m$^{-3}$) | N.A. | -0.49* | -0.08 | -0.16 | 0.14 | - |
| Calcium (ng m$^{-3}$) | N.A. | 0.01 | 0.14 | 0.15 | 0.26 | - |
| Nitrate (ng m$^{-3}$) | N.A. | -0.21 | -0.04 | -0.05 | 0.15 | - |
| Sulphate (ng m$^{-3}$) | N.A. | -0.39* | -0.12 | -0.15 | 0.13 | - |
| MSA (ng m$^{-3}$) | N.A. | 0.33* | 0.58** | 0.36* | 0.43* | - |

[1] $r$ in bold text represents results with statistical significance (p < 0.05); $r$ with * denote moderate correlation (0.3 < |$r$| < 0.5), and with ** indicate strong correlation (|$r$| > 0.5).

[2] "N.A." indicates that $r$ cannot be calculated due to the limited data pairs. "-" signifies the coupled variables, i.e., $N_{INP}$ at measured temperature and variables shown in the first column in Table 1 should not be correlated with each other due to the violation of e.g., measurement locations, size cut-off ranges between instruments.





Regarding the correlations between $N_{\mathrm{INP}}$ and size-resolved aerosol concentrations, it should be noted that increasingly higher correlations of $N_{\mathrm{INP}}$ were found with aerosol concentrations with increasingly larger size ranges at most nucleation temperatures, revealing the fact of the increasing contribution of larger-sized particles to INP populations. Nevertheless, the

overall absence of strong correlations is presented, which is in agreement with some previous studies (e.g., Li et al., 2022; Paramonov et al., 2020; Lacher et al., 2018). The absence of a notable correlation was also observed between $N_{\mathrm{INP}}$ and supermicron aerosol particles, despite previous findings (e.g., DeMott et al., 2015; Mason et al., 2016) suggesting them to constitute the major fraction of the observed INP population. One important reason is that INPs are only a small subgroup of ambient aerosol particles. As the time series of the activated fraction of INPs shown in Fig. 5, during the entire campaign,

approximately 1 out of $10^5$ ambient aerosol particles on average acted as INPs at an activation temperature of -15 °C. Therefore, a minor fluctuation of IN inactive components in the total aerosol populations, e.g., via a sudden increase of coarse-mode sea salt, would mask the correlations. Additionally, the long-range transport of INPs from mid- and high-latitudes in the upper troposphere, which is considered to be another important source of INPs in the remote Arctic (e.g., Porter et al., 2022; Schmale et al., 2021; Wex et al., 2019), may alter the population of INPs towards smaller sizes due to the size-dependent deposition

processes during atmospheric transport (Lacher et al., 2018). Similarly, strong correlations between $N_{\mathrm{INP}}$ and available surface area concentrations (i.e., $S_{>0.5}$ and $S_{<2.5}$ for correlating $N_{\mathrm{INP}}$ sampled by impinger and HINC, respectively) were not found.

Strikingly, the correlations between $N_{\mathrm{INP}}$ and fluorescent particle number concentrations, which have been used as a proxy for identifying bioaerosol (e.g., Toprak and Schnaiter, 2013; Savage et al., 2017) were relatively strong with statistical significance compared to other observed variables. This observation was particularly true towards warm nucleation temperatures, consistent

with our inference from our heat test results shown in Section 3.1. It is therefore reasonable to conclude that when the surface was free from snow and ice (as it was during most of the time during our measured period in the high Arctic), highly IN-active bioaerosols originating from the terrestrial and marine environments could act as dominant local INP sources. Additionally concerning $e\mathrm{BC}$, the overall weak to no correlations with $N_{\mathrm{INP}}$ at all investigated temperatures suggest negligible contributions from $e\mathrm{BC}$, which is in agreement with the findings that BC is not an effective INP in the immersion freezing mode in the MPC

temperature regime (-38 °C < $T$ < 0 °C), both from field (Paramonov et al., 2020; Lacher et al., 2018; Kupiszewski et al., 2016) and from laboratory studies (Kanji et al., 2020; Mahrt et al., 2018; Chou et al., 2013; Friedman et al., 2011).

Aside from the characterization of physical aerosol properties, we also investigated the relationships of meteorological variables with $N_{\mathrm{INP}}$. Schneider et al. (2021) improved the predictability of $N_{\mathrm{INP}}$ by using ambient temperature as a proxy for seasonal variations of INP abundance. However, a correlation between $N_{\mathrm{INP}}$ and ambient temperature ($T_{\mathrm{env}}$) was not observed

in this work. Similarly, ambient relative humidity ($RH_{\mathrm{env}}$) and pressure ($p_{\mathrm{env}}$) were not (or weakly) correlated with $N_{\mathrm{INP}}$. In contrast, moderate to strong correlations were found between $N_{\mathrm{INP}}$ ($T$ > -30 °C) and ground-level wind speed ($ws$), suggesting transport and advection of INPs within the sampling period. The concentration of SSA depends strongly on wind-induced wave breaking and bubble bursting (Lewis et al., 2004; Moallemi et al., 2021), which could enhance the local INP sources via, e.g., increased re-suspension of blowing dust and/or boosted SSA. Weak to moderate correlations were observed between $N_{\mathrm{INP}}$ and

ground-level wind direction ($wd$), suggesting that abrupt increases of local emissions from certain directions did not contribute to specific INP sources over a relatively long time span. Particularly, the highest $N_{\mathrm{INP}}$ was approximately associated with the





wind direction from the northern side, where there was an ice-free ocean during the entire measurement season, indicating the local marine SSA contributed to the INP concentrations. Given that sea salt is not an active INP in immersion mode, it is reasonable to conclude that the organic or biogenic components of SSA attribute to $N_{INP}$ as supported by the heat treatment
results and high correlations with fluorescent particle number concentrations.

The lower part in Table 1 shows the correlations between $N_{INP}$ derived from $PM_{10}$ filter samples and their ionic composition. Due to the limitation that the ion chromatography samples had a time resolution of two days, the number of samples was insufficient compared to other online measurement variables, resulting in the inadequate significance of correlations to $N_{INP}$. Nevertheless, moderate to strong correlations between $N_{INP}$ and the concentration of methanesulfonic acid (MSA) were ob-
served for most measured temperatures. MSA is oxidized from dimethylsulfide (DMS), originating from emissions by oceanic phytoplankton. Despite MSA itself does not act as INP, it is a unique indicator for tracing marine biological productivity since MSA has no other natural sources (Becagli et al., 2019). Therefore, the presented correlations demonstrate an association between INP abundance and local marine biological activity during our measured period. No correlations were manifested between $N_{INP}$ and other trace ions, including sodium, ammonium, calcium, nitrate and sulfate.

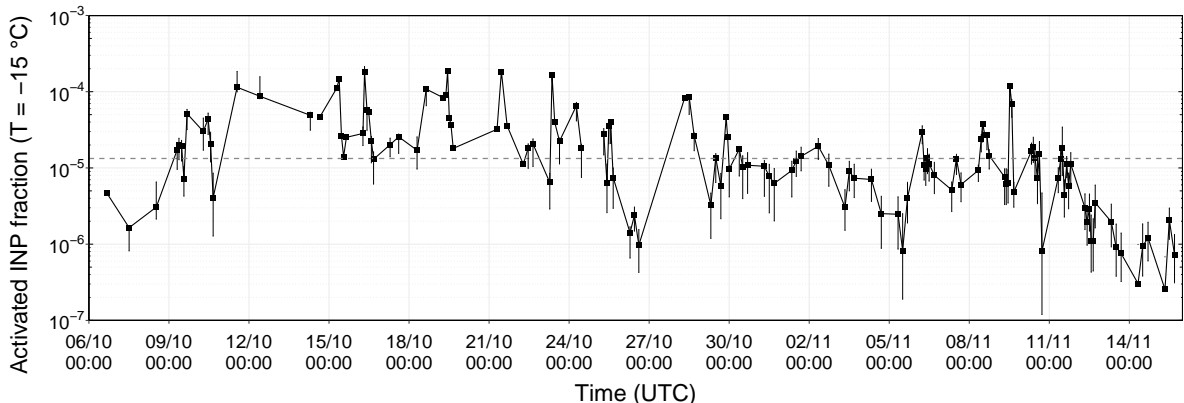

**Figure 5.** Time series for activated INP fraction at $T$ = -15 °C during autumn 2019 campaign in Ny-Ålesund. The activated INP fraction is the ratio between $N_{INP}$ measured from impinger samples and $n_{>0.5}$ measured from the APS. Vertical extensions represent the uncertainty range within 95 % confidence intervals. The dashed horizontal line indicates the median activated fraction over the entire campaign.

### 3.3 Special case studies

We further focussed on short-term cases, characterized by a broad range of $N_{INP}$, aerosol physicochemical properties, meteorological conditions and air mass origins. Figure 6 highlights the selected cases from INP samples representative of high, moderate and low INP concentrations during the period of highlighted heat-resistant high-$N_{INP}$ sample shown in Fig. 4 (c). A broader range of $N_{INP}$ were observed in impinger and $PM_{10}$ samples with finer temporal resolution of 1 hour and 8 hours,
respectively, revealing that the peak cases could explain the INP abundance in the corresponding 4-day WT-CRAFT sample. More evidence will be provided in the following subsections to elaborate on INP and aerosol features.





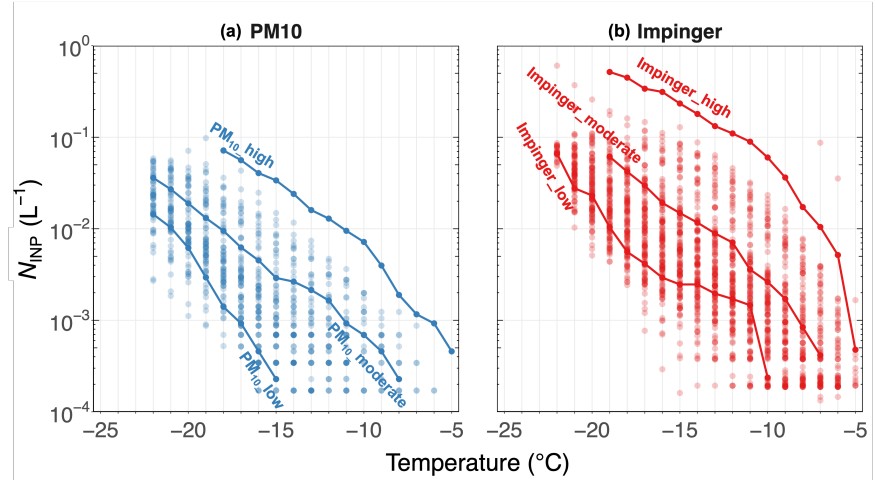

**Figure 6.** A selection of typical INP spectra (highlighted lines) labeled with high, moderate and low $N_{INP}$ for case studies for (a) $PM_{10}$ filter sampling from 19/10 00:00, 21/10 16:00 and 23/10 00:00, respectively; and for (b) impinger samples sampling from 21/10 11:05, 24/10 06:34 and 25/10 15:40, respectively. The dots in the background represent all measurements from offline techniques during the 6-week field measurement.

### 3.3.1 Time series

Figure 7 shows the time series during the period of selected impinger samples. All samples exhibited relative heat sensitivity, i.e., a reduction of INP concentrations was observed at $T$ = -15 °C (Fig. 7 [a]) for all selected cases (not detectable for some samples with values below the detection limit of DRINCZ). In addition, a high fluorescent particle population in the impinger_high case, (Fig. 7 [b]) indicates the probable existence of biological particles, which could serve as INPs at the investigated temperatures. However, for the impinger_low case, we still observed moderate levels of fluorescent particle concentrations, possibly due to the inclusion of the non-IN-active biological or abiotic fluorescent particles. Wind conditions can impact the local aerosolization process as shown in Fig. 7 (c). The time window for the impinger_high case was dominated by northerly winds coming from the direction of the ocean (see Figs. 1 [b] and 7 [c]), associated with higher wind speeds compared to the moderate and low INP cases. INP enrichment in the impinger_high sample could be attributed to promoted local SSA that were IN active, likely originating from biological production and aerosolization of the marine biota (Inoue et al., 2021). The size-resolved time series in Fig. 7 (d) coincided with the INP fluctuations, i.e., a rising $N_{INP}$ tendency with increasing coarse-mode particle number concentrations ($n_{>0.5}$) and total surface area concentrations ($S_{tot}$). This finding was consistent with previous ground-based observations that coarse-mode particles can strongly correlate with ambient INP population (e.g., DeMott et al., 2015, Mason et al., 2016). Concerning the moderate INP case, an abrupt rise in fine particle concentrations was observed, which had small contributions to the total surface area concentration and thus INP abundance. The low INP case was an exception regarding $n_{>0.5}$ and $S_{tot}$, which exhibited moderate levels of $n_{>0.5}$ and $S_{tot}$ but had the lowest $N_{INP}$ among the investigated cases.

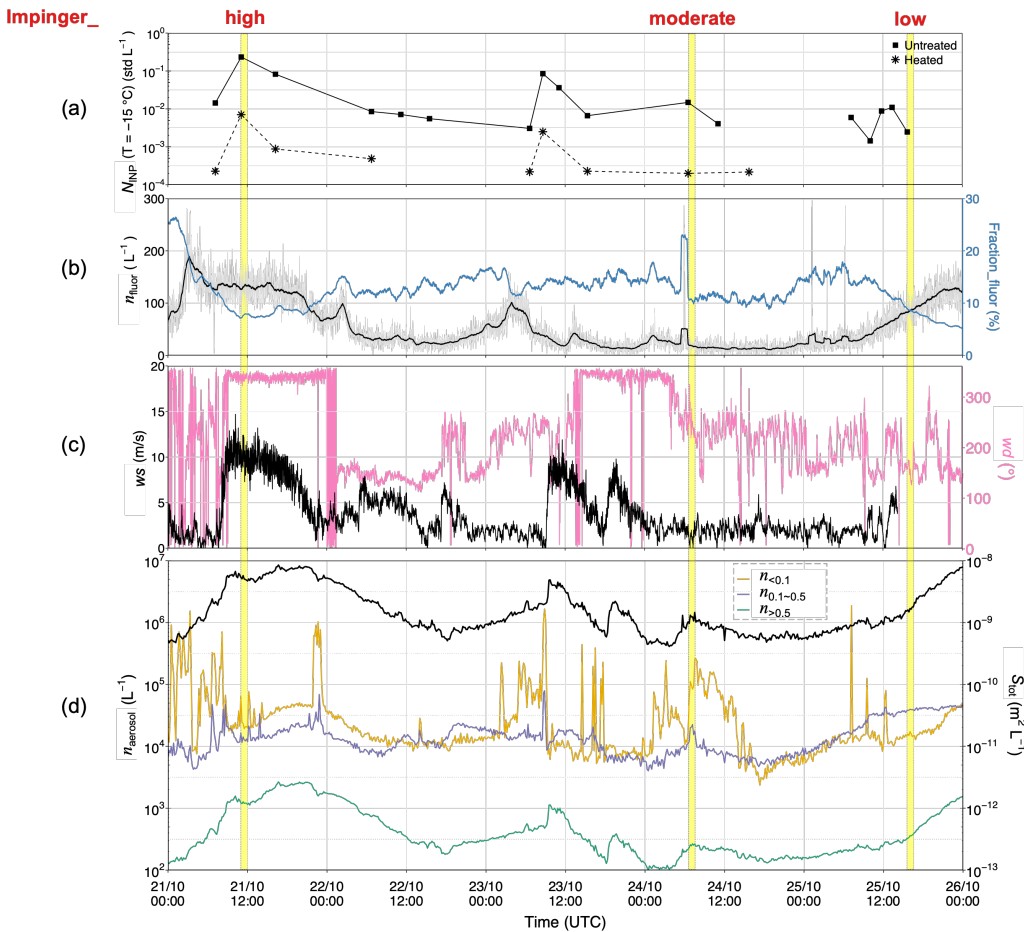

**Figure 7.** Time series in October 2019 during the window of selected impinger sample cases for (a) $N_{INP}$ at $T = -15$ °C (the entire temperature spectra can be found in Fig. 6 (b)), (b) fluorescent particle number concentrations ($n_{fluor}$, original 1-min data in light gray, and 30-min average in black) and fractions (Fraction_fluor), (c) wind speed ($ws$) and direction ($wd$) and (d) aerosol number concentration ($n_{aerosol}$) with different size ranges and total surface area concentration ($S_{tot}$) calculated for relevant sizes for impinger samples.

Interestingly, we noticed similar $N_{INP}$ - wind pattern for selected $PM_{10}$ samples (Fig. 8) from GVB that high and moderate $N_{INP}$ tended to be associated with high-speed winds with maritime origin. However, the size-resolved particle number and surface area concentrations were not good predictors for $N_{INP}$. The selected high, moderate and low $PM_{10}$ samples showed comparable $n_{>0.5}$ and $S_{tot}$ values (See Appendix A for the full range of particle size distribution). The possible reason is that aerosols sampled at the GVB observatory experienced more dilution and mixing due to the increased distance from the ocean.

Additionally, as discussed in Section 3.2, the coarse time resolution of $PM_{10}$ samples could mask the instantaneous enhanced INP loading by averaging it out over the 8-hour sampling period. Nevertheless, the high loading of particles with sizes larger than approximately 2.5 µm could contribute to the high INP case of the $PM_{10}$ sample (shown in Fig. A1).

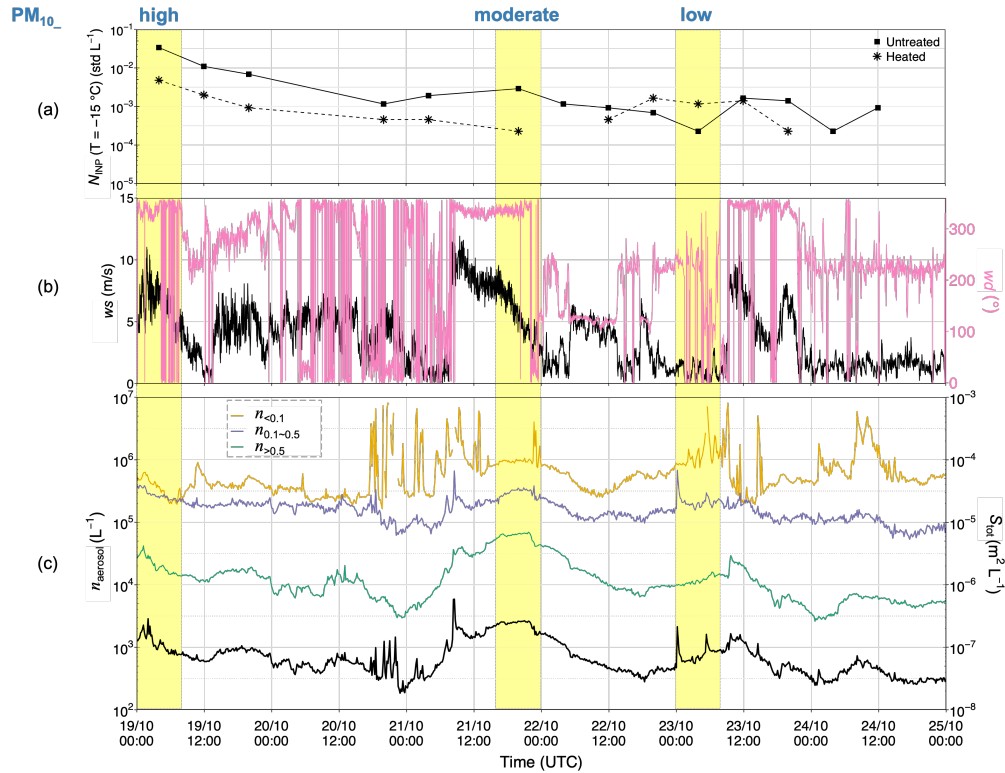

**Figure 8.** Time series in October 2019 during the window of selected $PM_{10}$ sample cases for (a) $N_{INP}$ at $T$ = -15 °C (the entire temperature spectra can be found in Fig. 6 (a)), (b) wind speed ($ws$) and direction ($wd$) and (d) aerosol number concentration ($n_{aerosol}$) with different size ranges and total surface area concentration ($S_{tot}$) calculated for relevant sizes for $PM_{10}$ samples.

### 3.3.2 Physicochemical characterization of selected samples

Similar to Fig. 4, Figure 9 displays individual INP spectra exposed to heat treatment or storage conditions for selected impinger
and $PM_{10}$ cases overlapping the period of the highlighted WT-CRAFT sample shown in Fig. 4 (c). Most selected samples were heat-labile in terms of $N_{INP}$, especially for the impinger_high sample at temperatures higher than -10 °C, revealing potential biogenic INP sources. $PM_{10}$_low was an exception, showing heat-resistant aerosol compositions, consistent with the heating results of the corresponding WT-CRAFT sample. A likely reason could be that with a much coarser resolution of the 4-day interval for the WT-CRAFT sample, the heat-labile INPs were overshadowed by the heat-resistant INPs, therefore showing
no degradation in $N_{INP}$ (Figure 4 [c]). This motivates the necessity of finer temporal resolution of INP measurements in the field study, despite the increased labor intensity. Additionally, the level of $N_{INP}$ after heat treatment (in red) also approximately followed the INP abundance classification (i.e., high-moderate-low), suggesting that the relative abundance of relatively heat-resistant INPs, i.e., mineral dust particles, although generally low in the background, may still explain the difference of $N_{INP}$ in the selected samples.

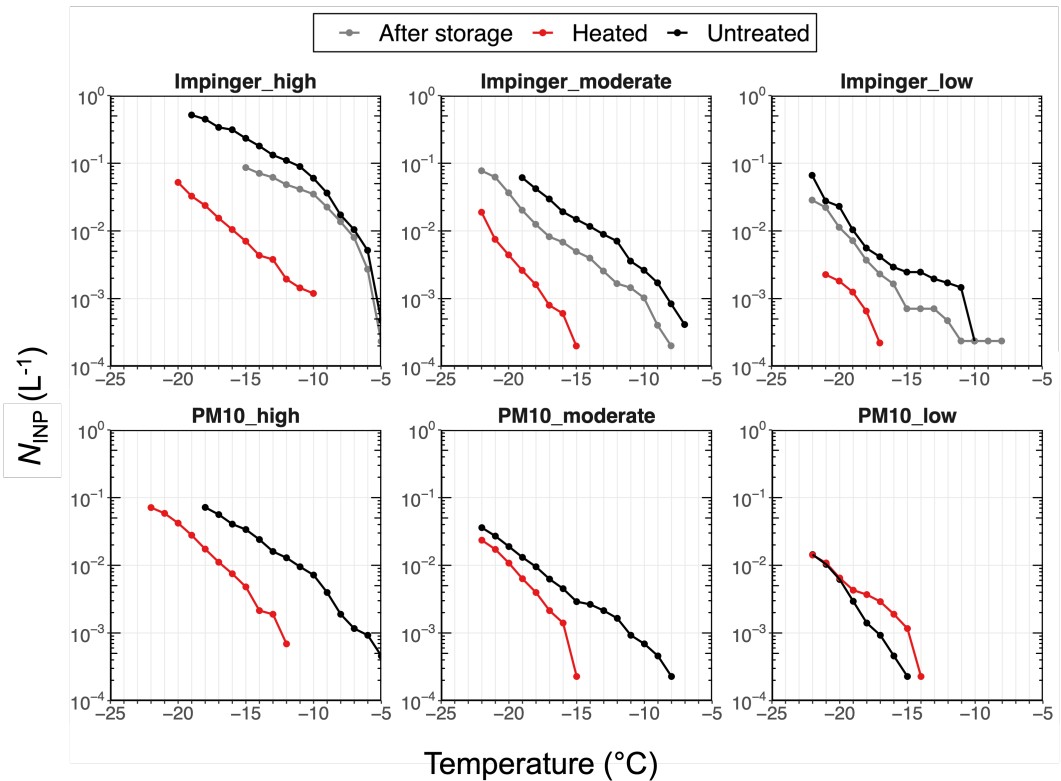

**Figure 9.** Selected INP spectra for untreated, heated and storage conditions labeled high, moderate and low $N_{INP}$ for case studies for impinger and PM$_{10}$ samples. Note that the time stamps for the same $N_{INP}$ labels regarding the impinger and PM$_{10}$ samples are different. High INP cases: 11:05-12:05 UTC 21/10/2019 and 00:00-08:00 UTC 19/10/2019, moderate INP cases: 06:34-07:34 UTC 24/10/2019 and 16:00-24:00 UTC 21/10/2019, and low INP cases: 15:40-16:40 UTC 25/10/2019 and 00:00-08:00 UTC 23/10/2019 for impinger and PM$_{10}$ samples, respectively.

We evaluated the chemical compositions of the representative subset of droplet residual (impinger) and particular matter (PM$_{10}$) filter samples labeled with high, moderate and low $N_{INP}$ from offline IN measurements, in order to understand the diverse chemical compositions and sources (Figure. 10). Note the slight differences for classifications in impinger droplet residual and PM$_{10}$ samples due to different probed elements (see detailed atomic fractions in Appendix C) and different sampling substrates. C+N, P (Phosphorus), metal, dust and salt are major compositions of impinger droplet residual samples

(Figure 10 (a), (b) and (c)). The compositional diversity suggests that the sea salts and minerals sampled in our study could be aged, and mixed, but we cannot comment on the mixing state (internal vs. external) because all aerosols were sampled into the same liquid sample, allowing for post-sampling mixing. Abundant carbonaceous organics (C+N) were detected in the impinger_moderate sample, possibly released post to marine biological productions as suggested by the fluorescent signals shown in Fig. 7 (b). Additionally, although higher dust and lower salt contents were detected in impinger_low samples, the





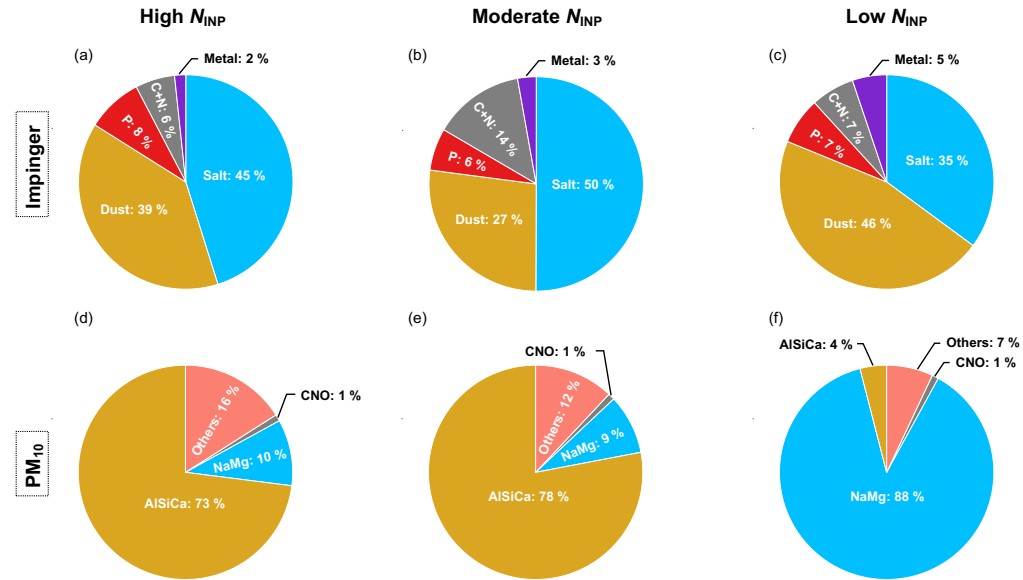

**Figure 10.** Normalized chemical composition fractions from SEM-EDX for selected (a), (b) and (c) impinger samples (droplet residuals), (d), (e) and (f) PM$_{10}$ samples. Note that the categorization based on elemental compositions was slightly different for selected impinger (Cheng et al., 2022) and PM$_{10}$ (Hiranuma et al., 2013) samples. The group C+N (CNO) include particles containing only carbon and nitrogen (and oxygen), mainly representative of (oxygenated) carbonaceous particles. The salt and NaMg classes are identical, composed of particles containing sodium and magnesium salts typically indicative of sea salt. The dust class is similar to AlSiCa except Al was not included in the categorization of dust in impinger droplet residual samples that were collected on aluminum substrates.

INP concentration was low. A possible reason could be the suppression of ice nucleation activity of dust particles when aged in an aqueous environment (Kumar et al., 2018) in the impinger samples. Table 2 summarizes the detection of organic functional groups (bonds) for selected impinger droplet residual samples using Raman microspectroscopy. Organic-rich functional groups were identified in the high and moderate INP sample, associated with possible co-emission with marine biogenic aerosols as previously discussed. In contrast, no organic functional groups were identified in the impinger_low sample.

Concerning PM$_{10}$ samples, greater dust (AlSiCa) content was associated with samples with higher $N_{INP}$, demonstrating major terrestrial sources. In contrast, more sea salt was pronounced in the low-$N_{INP}$ sample. It remains unknown whether the abundance of INP of maritime origin is due to the presence of other marine constituents occasionally co-emitted with sea salt particles, such as sulfates or organic carbon, in the elevated particles during periods of marine biological activity.

To summarize, our single particle microspectroscopy results justify (1) the aerosol particle composition is not equivalent to 475 the INP composition; (2) the variation in aerosol composition can infer the particle source and air mass history but cannot be a direct indicator of the INP abundance; and (3) quantitatively small organic compounds can substantially influence the INP prosperities at least in our studied samples collected in the Arctic.





**Table 2.** Summary of the number (#) of functional groups/chemical bonds detected via the Raman spectroscopy for selected impinger droplet residual samples.

| Functional group/bond | Impinger_high (#) | Impinger_moderate (#) | Impinger_low (#) |
|---|---|---|---|
| Organic | 13 | 12 | < LOD* |
| Metal oxide(s)** | 10 | 2 | < LOD |
| Nitrogen bond(s) | 3 | 5 | < LOD |
| Aromatic ring(s) | 4 | 3 | < LOD |
| Total NO. of spectra | 13 | 12 | < LOD |

*Limit of detection.

**Possible interference from the aluminum foil that was used as sample substrate.

### 3.3.3 HYSPLIT backward trajectories for selected case studies

Backward trajectory analysis was conducted to assess the origin of sampled air masses and to identify potential long-range

sources of the measured INPs in the Arctic coastal region in Ny-Ålesund. Figure 11 shows the air masses during the sampling period of the impinger_high case originating from the coastal regions in the vicinity of Greenland, indicating possible influences from long-range transport of terrestrial sources, which qualitatively justifies the inclusion of some dust and organics. Similarly, rich organic and sea salt particles identified in the impinger_moderate sample could be attributed to the potential impacts from lower latitudes, where the residence time of air masses was much longer over the ice-free Barents and Kara Seas. A clear

exception of air mass history was observed for the impinger_low sample when the air circulation within the Arctic circle was mostly over the ice pack or locally over Spitzbergen, which explains the low concentrations of INPs, organics and aerosol particles.

Air mass trajectories during the sampling time of $PM_{10}$ filters (Figure 12) show that predominantly the trajectories were originated from western and northern Greenland in the high and moderate cases (Figures 12 [a] and [b]), which coincided with

relatively abundant INPs and high-latitude dust. These consistencies in high INPs and dust from Greenland for both $PM_{10}$ and impinger samples suggest that long-range transported dust from high latitudes (e.g., Greenland) could play an important role in the INP population in the remote Arctic regions.





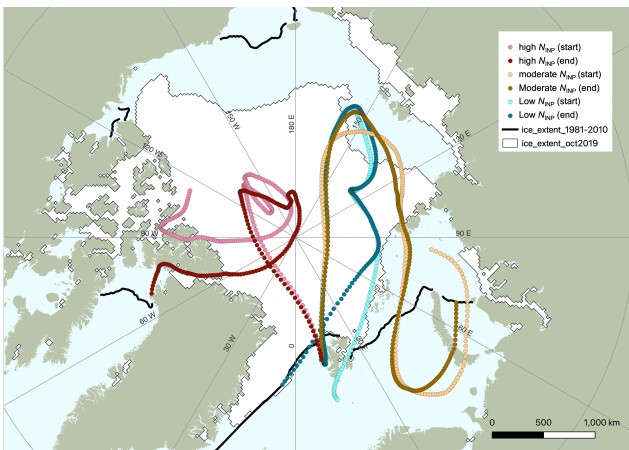

**Figure 11.** HYSPLIT backward trajectories over 10 days, starting at the sampling location at 5 m a.g.l height every hour within the sampling period (2 trajectories per sample), for the selected impinger INP case studies throughout the campaign. High INP case: 11:05 to 12:05 UTC 21/10/2019, moderate INP case: 06:34 to 07:34 UTC 24/10/2019 and low INP case: 15:40 to 16:40 UTC 25/10/2019.

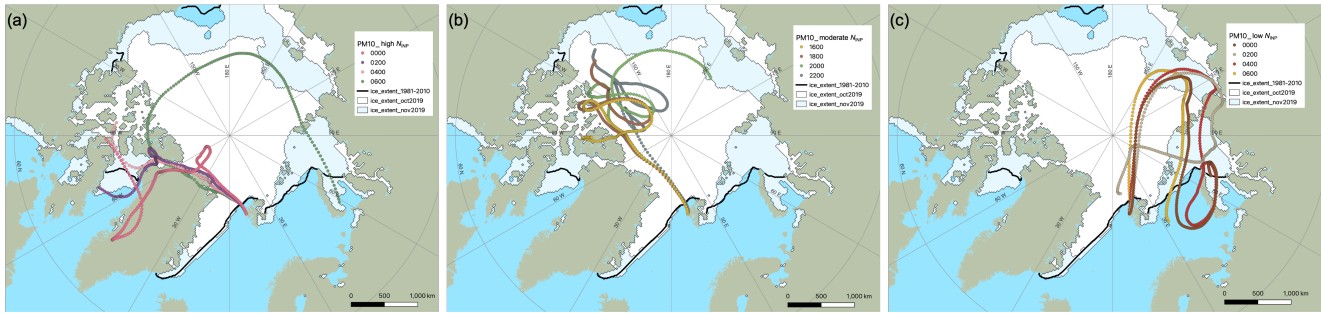

**Figure 12.** HYSPLIT backward trajectories over 10 days, starting at the sampling location at 5 m a.g.l height every 2 hours (3 trajectories per sample), for the selected PM$_{10}$ INP case studies throughout the campaign. High INP case from 00:00 to 08:00 UTC 19/10/2019, moderate INP case from 16:00 to 24:00 UTC 21/10/2019 and low INP case from 00:00 to 08:00 UTC 23/10/2019.

## 4 Summary and Conclusions

This study presents the measurement results of ambient INP concentrations and related aerosol properties during the NASCENT campaign in Ny-Ålesund, Svalbard in October-November 2019. A combination of online and offline INP measurement techniques were applied in order to obtain a broad range of $N_{INP}$-$T$ spectra and to understand the spatiotemporal variability of INPs from fine to coarse temporal resolutions.

Overall, we observed that $N_{INP}$ was approximately two orders of magnitudes lower compared to the global average (Petters and Wright, 2015) and was generally in good agreement with $N_{INP}$ from previous studies in Ny-Ålesund. We showed that the





majority of offline samples experienced a degradation in $N_{INP}$ upon heat treatment, particularly towards warm temperatures (i.e., $T > $-15 °C), indicating the likely presence of proteinaceous or biogenic INPs. Correlation results linking aerosol properties to $N_{INP}$ exhibited weak associations between $N_{INP}$ and coarse-mode particles, despite their importance being highlighted by many previous studies. The averaging effect over relatively coarse-resolution data, scarcity of INPs and possible long-range modification processes were proposed as potential causes. Relatively strong correlations were found between $N_{INP}$ and particle

fluorescence, suggesting that highly IN-active bioaerosols, which may be inherently related to particles large in size during the snow and ice-free season could serve as dominant local INP sources in the remote Arctic. The relationship between INP abundance and ocean-oriented high-speed wind and MSA concentrations further supports that $N_{INP}$ could be contributed from locally enhanced SSA of biological origin.

     Moreover, case studies with scenarios for a typical range of $N_{INP}$ collected closer to the coast were presented (i.e., impinger

samples at the aerosol container). The high $N_{INP}$ case was associated with strong heat-lability, fluorescence, high wind speed of maritime origin, elevated concentration of coarse-mode particles and surface area and abundant organics. Chemical composition analyses reveal that the diversity in aerosol composition did not have a substantial impact on INP abundance, which could be a good future motivation to investigate the composition of cloud/ice residual and mixing state of aerosols and their associated impact on INP population and properties. Backward trajectories demonstrated possible high-latitude dust sources

from long-range transport (e.g., coastal Greenland) that could be responsible for the INP enrichment. In contrast, for low-$N_{INP}$ cases, most of air mass history was experienced over the ice-pack zone. This research increased the data coverage of INP measurements in the remote Arctic and provided comprehensive analyses of the INP physicochemical properties and potential sources. Further studies are needed in different seasons to elucidate the annual sources of INPs in the Arctic.

## Appendix A: Particle size distribution of case studies

Figure A1 provides full spectra of averaged particle size distribution within the time window for selected case studies presented in Figs. 7 and 8. The INPs collected by the impinger include only particles with aerodynamic diameters larger than 500 nm. Therefore, the high coarse-mode particle concentrations contributed to high INP case, but a few particles with diameters larger than approximately 8 μm, which were likely representative for instantaneously enhanced local sources, should explain the higher INP concentrations in impinger_moderate compared to impinger_low sample. Concerning $PM_{10}$ samples, the filters

collected all particle sizes below 10 μm. The high INP concentration in PM10_high sample could be attributed to the high particle concentrations with $D_p > $ ca. 2.5 μm. However, compared to the PM10_low sample, particle loading in the PM10_moderate sample was comparable when $D_p > 2.5$ μm, and were rather dominated by those under ca. 2.5 μm.





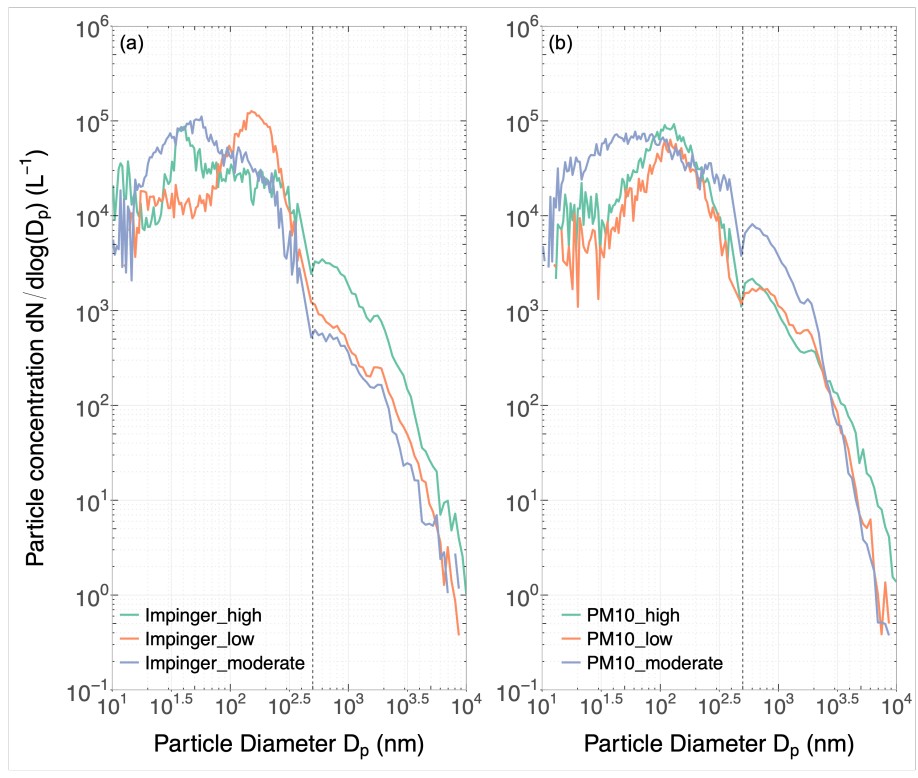

**Figure A1.** Average particle size distribution of selected cases of (a) impinger; and (b) PM$_{10}$ sample. The vertical dashed lines indicate D$_p$ = 500 nm, approximately the size boundary of SMPS and APS measurement.

**Appendix B: SEM image of selected samples**





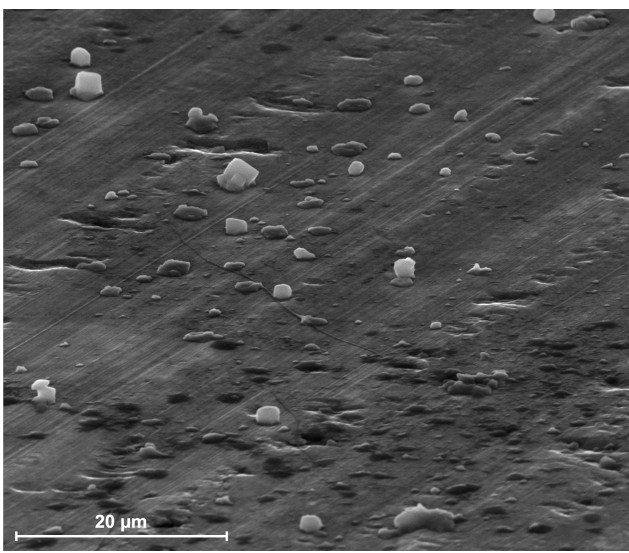

**Figure B1.** SEM image of impinger_high samples (tilted view).

## Appendix C: Atomic fraction for selected samples

**Table C1.** Atomic fraction of droplet residuals for selected impinger samples from CCSEM-EDX results.

| Atoms | Impinger_high (%) | Impinger_moderate (%) | Impinger_low (%) |
|-------|-------------------|-----------------------|-------------------|
| C  | 9.2±10.3   | 24.2±13.8  | 10.8±12.0  |
| N  | 0.2±1.1    | 0.2±1.1    | 0.1±0.7    |
| O  | 13.7±7.4   | 18.1±10.2  | 13.7±9.3   |
| Na | 20.9±22.9  | 9.2±11.0   | 7.8±8.5    |
| Mg | 14.0±7.1   | 18.4±7.1   | 15.3±9.1   |
| Si | 9.4±7.7    | 6.8±5.8    | 11.2±8.2   |
| P  | 0.2±0.6    | 0.2±0.7    | 0.2±0.7    |
| S  | 0.7±1.2    | 0.9±1.5    | 0.7±1.3    |
| Ca | 2.0±1.7    | 2.0±3.0    | 2.0±2.0    |
| Mn | 2.6±1.8    | 2.8±2.2    | 3.1±2.0    |
| Fe | 23.0±22.8  | 11.5±13.4  | 29.9±24.4  |
| Zn | 4.2±2.5    | 5.9±2.9    | 5.2±3.2    |



**Table C2.** Atomic fraction of particulate residuals for selected $PM_{10}$ samples from SEM-EDX results.

| Atoms | $PM_{10}$_high (%) | $PM_{10}$_moderate (%) | $PM_{10}$_low (%) |
|---|---|---|---|
| C | 70.9±7.4 | 72.7±6.0 | 83.3±7.5 |
| N | 0.0±0.0 | 0.0±0.0 | 16.3±4.4 |
| O | 26.7±5.8 | 25.3±5.2 | 13.3±6.4 |
| Na | 0.6±0.4 | 0.5±0.5 | 1.7±1.1 |
| Mg | 1.3±1.6 | 0.6±0.4 | 0.5±0.3 |
| Al | 0.8±0.6 | 0.7±0.5 | 0.6±0.6 |
| Si | 1.0±1.1 | 1.0±0.8 | 1.6±2.4 |
| P | 0.1±0.0 | 0.1±0.1 | 0.0±0.0 |
| S | 0.3±0.3 | 0.5±0.5 | 0.2±0.2 |
| Cl | 0.2±0.2 | 0.1±0.2 | 0.8±0.6 |
| K | 0.2±0.2 | 0.1±0.1 | 0.1±0.0 |
| Ca | 0.5±0.5 | 0.4±0.5 | 0.1±0.1 |
| Mn | 0.2±0.2 | 0.1±0.0 | 0.0±0.0 |
| Fe | 0.7±0.7 | 0.4±0.5 | 0.3±0.1 |
| Zn | 0.1±0.0 | 0.1±0.0 | 0.1±0.0 |

*Data availability.* The data presented in this study are available at XX. Note by authors: data will be uploaded upon acceptance of publication.

*Author contributions.* GL performed the data analysis, produced figures and wrote the original manuscript draft with contributions from all co-authors. GL and JW performed the INP and aerosol sampling and measurements and post-sample processing and data analyses. EW conducted and led WT-CRAFT, SEM-EDX and back trajectory analyses and interpreted the results. ZC and SC were responsible for 535 CCSEM-EDX analysis, data interpretation and idea input for the study. EW, AF, SB and NH conducted the Raman spectroscopy analysis. JH was involved in organizing the field study and provide feedback on the manuscript. GM and AN provided the WIBS instrument and pre-processed WIBS data. RT provided the ion chromatography data. MM provided meteorological data from the climate change tower next to the GVB observatory. NH conceived the idea of the study and contributed to the data interpretation, idea input and manuscript feedback. UL was involved in providing feedback on the research ideas and manuscript. ZAK supervised the research processes during campaign planning, 540 experiments, data interpretation and manuscript writing.

*Competing interests.* The authors declare that no competing interests are present.



*Acknowledgements.* GL and ZAK acknowledge that this project has been made possible by a grant from the Swiss Polar Institute, Dr. Frederik Paulsen. NH acknowledges the National Science Foundation under grant no. 1941317. The authors thank the staff of the CNR Arctic Station for their support. The WT-CRAFT team acknowledges Hemanth S.K. Vepuri for his support on measurements and analyses. EW and NH thank Jacob Hurst for his support on SEM-EDX measurements and analyses. ZC and SC acknowledge support from the Environmental Molecular Sciences Laboratory, a DOE Office of Science User Facility sponsored by the Biological and Environmental Research program under Contract no. DE-AC05-76RL01830. We acknowledge all those involved in the fieldwork associated with the NASCENT project, including technical support from Dr. Michael Rösch, Dr. Robert O. David, and from the AWIPEV and Norwegian Polar Institute. We would like to thank Dr. Sho Ohata and Dr. Yutaka Tobo for sharing their research data on black carbon measurement, and Stefania Gilardoni for supporting the PSAP data. We want to express our gratitude to Dr. Jie Chen and Dr. Nadia Shardt for the insightful discussions.





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
