# Peer review of "Physicochemical Characterization and Source Apportionment of Arctic Ice Nucleating Particles Observed in Ny-Ålesund in Autumn 2019"

_Atmospheric Chemistry and Physics, 2023_

## Referee Comment (RC1)

This study presents the measurement results of ambient INP concentrations and related aerosol properties during the NASCENT campaign in Ny-Ålesund, Svalbard in October-November 2019.The paper describes the complexity of ice nucleating particles (INP) in Arctic coastal environment. Physicochemical parameters have been analyzed and tested and are ice nucleation temperature, heat-lability, and fluorescence activity, which were correlated with environmental parameters such as wind speed, wind direction, temperature, and snow coverage. The results are highly interesting and assume high-latitude dust sources from long-range transport that could be responsible for INP enrichment. This paper should be published after some minor changes:

2.1 measurement location: Please provide already here the GPS coordinates for the aerosol container and the GVB station.

Instead of figure 1b, a map would be more helpful showing the distances and a wind rose including wind speed and wind direction frequencies.

Figure 2 is valuable for the interested reader but might be shifted to the appendix.

2.2 INP sampling techniques: You might discuss the different sizes and number of droplets been investigated with the three techniques (WT-CRAFT, HINC and DRINCS).  What is the impact of both parameters on the error bars, the homogeneous ice nucleation temperature and the limit of detection? Please provide a discussion which makes these differences and their impact on the results more transparent for the reader.

2.3. Heat treatment: As already discussed by many authors, the impact of heat treatment is an ambiguous procedure (please quote the respective literature), e.g. some low-molecular INM from pollen can be rather heat stable ($T_{on}$<-15°C), while high-molecular INM agglomerates from the same source are losing INA due to heat treatment. A more reliable technique is digestion of the INP/INMs with enzymes, which will give evidence for the presence of proteins. Treating ice nucleation active samples with enzymes (e.g. Kozloff et al., 1991; Burkart et al. 2021, Pummer et al., 2012; Felgitsch et al., 2019), chaotropic reagents (e.g. Pummer et al., 2012; Fröhlich-Nowoisky et al., 2015), or a strong oxidizer (e.g. $H_2O_2$; Gute et al., 2020) to investigate the nature of ice nuclei has been performed in the past.

3.1 INP concentrations: In figure 4, it is interesting to note that in the range -5° to -15°C the INA decreases a lot, which might be interpreted as proteinaceous aggregates (Seifried et al. 2023). In figure 4c the legend with full circles is missing. Eventually, the authors might use different symbols related to the different colors. In figure 4c also error bars are missing.

In the text you mention a sensitivity of the proteins due to cryo-storage. Please provide explanation for this effect.

Table 1: The Pearson correlation coefficient is listed. Please, explain why a r>0.5 indicates strong correlation. How this category has been defined? Also, for $N_{INP}$(T=-6°C) / $n_{AC+ABC}$ the 0.63 should have two asterixis.

Figure 7 and 8: The labels on the axes are rather small.

Figure 9: Please us different symbols related to the different colors.

Table 2: What means "NO"?

Figure 11 and 12: The labels in the figures are much too small and are unreadable.

Figure B1: Please describe the figure in more detail. The particles have the typical shape of NaCl crystals. Please mark the spots in the SEM where the EDX has been recorded. If available an EDX mapping might be shown.

References

Burkart, J., Gratzl, J., Seifried, T. M., Bieber, P., and Grothe, H.: Isolation of subpollen particles (SPPs) of birch: SPPs are potential carriers of ice nucleating macromolecules, Biogeosciences, 18, 5751–5765, https://doi.org/10.5194/bg-18-5751-2021, 2021.

Felgitsch, L., Bichler, M., Burkart, J., Fiala, B., Häusler, T., Hitzenberger, R., and Grothe, H.: Heterogeneous Freezing of Liquid Suspensions Including Juices and Extracts from Berries and Leaves from Perennial Plants, Atmosphere-Basel, 10, 37, https://doi.org/10.3390/atmos10010037, 2019.

Fröhlich-Nowoisky, J., Hill, T. C. J., Pummer, B. G., Yordanova, P., Franc, G. D., and Pöschl, U.: Ice nucleation activity in the widespread soil fungus Mortierella alpina, Biogeosciences, 12, 1057–1071, https://doi.org/10.5194/bg-12-1057-2015, 2015.

Gute, E. and Abbatt, J. P.: Ice nucleating behavior of different tree pollen in the immersion mode, Atmos. Environ., 231, 117488, https://doi.org/10.1016/j.atmosenv.2020.117488, 2020.

Gute, E., David, R. O., Kanji, Z. A., and Abbatt, J. P.: Ice Nucleation Ability of Tree Pollen Altered by Atmospheric Processing, ACS Earth Space Chem., 4, 12, 2312–2319, 2020

Kozloff, L. M., Turner, M. A., and Arellano, F.: Formation of bacterial membrane ice-nucleating lipoglycoprotein complexes, J. Bacteriol., 173, 6528–6536, https://doi.org/10.1128/jb.173.20.6528-6536.1991, 1991.

Pummer, B. G., Bauer, H., Bernardi, J., Bleicher, S., and Grothe, H.: Suspendable macromolecules are responsible for ice nucleation activity of birch and conifer pollen, Atmos. Chem. Phys., 12, 2541–2550, https://doi.org/10.5194/acp-12-2541-2012, 2012.

Seifried, T.M.; Reyzek, F.; Bieber, P.; Grothe, H. Scots Pines (Pinus sylvestris) as Sources of Biological Ice-Nucleating Macromolecules (INMs). Atmosphere 2023, 14, 266. https://doi.org/10.3390/atmos14020266

---

## Author Comment (AC1)

**Referee comments 1**

**Review of "Physicochemical Characterization and Source Apportionment of Arctic Ice Nucleating Particles Observed in Ny-Ålesund in Autumn 2019" by Li et al., submitted to ACPD**

**This study presents the measurement results of ambient INP concentrations and related aerosol properties during the NASCENT campaign in Ny-Ålesund, Svalbard in October-November 2019.The paper describes the complexity of ice nucleating particles (INP) in Arctic coastal environment. Physicochemical parameters have been analyzed and tested and are ice nucleation temperature, heat-lability, and fluorescence activity, which were correlated with environmental parameters such as wind speed, wind direction, temperature, and snow coverage. The results are highly interesting and assume high-latitude dust sources from long-range transport that could be responsible for INP enrichment. This paper should be published after some minor changes:**

We thank referee 1 for the valuable feedback on our manuscript acp-2023-18. In response to the questions and suggestions, please find our answers and revisions listed below. **Referee comments are reproduced in bold** and author responses in normal font; *extracts from the original manuscript are presented in red italic* and *extracts from the revised manuscript in blue italic*.

**2.1 measurement location: Please provide already here the GPS coordinates for the aerosol container and the GVB station.**
We now add the GPS coordinates for the aerosol container and the GVB station in the revised manuscript (see lines 84-85). In addition, we state the direction and distance relative to the container see caption of Figure 1 in the revised manuscript.

**Instead of figure 1b, a map would be more helpful showing the distances and a wind rose including wind speed and wind direction frequencies.**
We agree and have now added distances and a wind rose for the relevant dates discussed in this paper (see Fig. 1b in the revised manuscript). In addition, we adjust the descriptions to *"Local sources of pollution have a limited influence on the measurement sites during the measurement period, given the predominant southeasterly wind at the aerosol container and prevailing southwesterly winds close to the GVB observatory station (see detailed wind pattern in Fig. 1 b)"* (see lines 85-87 in the revised manuscript).

**Figure 2 is valuable for the interested reader but might be shifted to the appendix.**
We appreciate this comment, but we decided to keep Fig. 2 in the revised manuscript because it is difficult to introduce the experimental setup in Section 2 without referring to a figure and we believe it is important for reproducing such an experiment in the future.

**2.2 INP sampling techniques: You might discuss the different sizes and number of droplets been investigated with the three techniques (WT-CRAFT, HINC and DRINCZ). What is the impact of both parameters on the error bars, the homogeneous ice nucleation temperature and the limit of detection? Please provide a discussion which makes these differences and their impact on the results more transparent for the reader.**
We agree with the reviewer's comment. As shown in Equation (1) in the revised manuscript, both droplet size and number impact the error bars and the limit of detection of the measurement. Additionally, the homogeneous freezing temperature is impacted by the droplet size, with smaller droplets requiring lower homogeneous freezing temperature (Koop and Murray, 2016). To make the comparison clearer, we add Table 1 in the revised manuscript, summarizing the features of different INP sampling and measurement techniques along with their LOD. The table is now discussed in lines 106-111 of the revised manuscript: *"To investigate the ambient INP concentrations in immersion-freezing mode, we used different INP sampling and measurement instruments introduced in the following subsections, which provide a large range of sampled particle sizes, time resolutions, freezing temperatures, and hence different INP detection limits (see Table 1).*

*In particular, the droplet-freezing techniques (see Sections 2.2.1 and 2.2.2) have different limits of detection (LOD) due to the different droplet sizes and numbers in the experimental setup."*

**2.3. Heat treatment: As already discussed by many authors, the impact of heat treatment is an ambiguous procedure (please quote the respective literature), e.g., some low-molecular INM from pollen can be rather heat stable ($T_{on}$ < -15 ℃), while high-molecular INM agglomerates from the same source are losing INA due to heat treatment. A more reliable technique is digestion of the INP/INMs with enzymes, which will give evidence for the presence of proteins. Treating ice nucleation active samples with enzymes (e.g. Kozloff et al., 1991; Burkart et al. 2021, Pummer et al., 2012; Felgitsch et al., 2019), chaotropic reagents (e.g. Pummer et al., 2012; Fröhlich-Nowoisky et al., 2015), or a strong oxidizer (e.g. $H_2O_2$, Gute et al., 2020) to investigate the nature of ice nuclei has been performed in the past.**

The following references (Hill et al., 2016 and Pummer et al., 2015) suggest using heat treatment to evaluate the contribution of heat-labile materials to the INP population, representing proteinaceous IN components that are degraded after heating. Heating is a qualitative approach to investigate the presence of heat-labile/biological components, although we notice the possible ambiguity of this method suggested by the referee. As suggested by the referee, more definitive approaches e.g., Burkart et al. (2021) treated sub-pollen particles with a protease to investigate the quantitative connection between the IN activity and protein concentration. Pummer et al. (2012) treated the most efficient pollen-INPs to identify the proteinaceous and non-proteinaceous compounds. In this work, we were not aiming at identifying the type of heat-labile/biological particles. Instead, our work focused on detecting the presence vs. absence of those particles by using the heating method. Additionally, a finer assessment of the types of biological nucleators in our samples was not possible due to the limited sample volume available from the campaign. We wanted to use the samples to investigate the storage impacts and further chemical and spectroscopic analysis that are presented here. We removed statements about the specificity of biological nucleators in the revised manuscript other than implying that they are heat-labile. We also include a statement in section 2.3 (lines 197-199) to clarify this point *"We note that such heat treatment could exclude lower molecular weight samples yet still imply that proteinaceous aggregates are present (Seifried et al. 2023). Thus, any effect of heat treatment on the INP concentration would be due to the contribution of heat-labile particles from biogenic sources."*

**3.1 INP concentrations: In figure 4, it is interesting to note that in the range -5 to -15 ℃ the INA decreases a lot, which might be interpreted as proteinaceous aggregates (Seifried et al. 2023). In figure 4c the legend with full circles is missing. Eventually, the authors might use different symbols related to the different colors. In figure 4c also error bars are missing.**

In the revised manuscript, we have now used different symbols corresponding to different colors. In addition, the error bars and the legend for different symbols have been added in Fig. 4c (which has been moved to Fig. 6c in the revised manuscript also inspired by the reviewer's comments for a better comparison, see below).

**In the text you mention a sensitivity of the proteins due to cryo-storage. Please provide explanation for this effect.**

Two reasons are suggested in Beall et al. (2020). We now add the following description in lines 352-356 of the revised manuscript *"The number of small organic INPs could be reduced due to aggregation when enriched solute becomes incorporated into the ice phase during storage. Additionally, as the solution phase is enriched during freezing, smaller INPs may be absorbed onto the surface of larger particles, thus resulting in the coalescence of the INPs (Beall et al., 2020). However, a clear mechanism for the INP losses after cryo-storage is not reported since they lack the identities of observed INPs."* However, additional reasons are responsible for the reduction in INP concentration for PM10 filter samples. We add the following reasoning sentences in lines 358-362 of the revised manuscript *"The above reasons, however, would not explain degradation in the $PM_{10}$ samples; as such, we believe that the lower $N_{INP}$ in the $PM_{10}$ samples is indicative of a size dependency since the impinger samples include particles larger than 10 $\mu m$ which are excluded in the $PM_{10}$ samples. This conclusion is also supported by the $N_{INP}$ from impinger being systematically higher than those from the $PM_{10}$ samples (see Fig. 3)."*

**Table 1: The Pearson correlation coefficient is listed. Please, explain why a r > 0.5 indicates strong correlation. How this category has been defined? Also, for $N_{INP}$ (T = -6 ℃) / $n_{AC+ABC}$ the 0.63 should have two asterixis.**

Thank you for pointing this out. The double asterix is added to $n_{AC+ABC}$ = 0.63. Regarding the strength of the Pearson correlation coefficient, there is no fixed definition of the level of correlation corresponding to a specific coefficient value. In addition to the authentic correlation, the absolute correlation coefficient also depends on the quality of the data set. Considering the similarity in the format of the data set obtained from aerosol field measurements, we took the classification of correlation strength indicated in Lacher et al. (2018); Paramonov et al. (2020) and Rinaldi et al. (2021). As such we base our threshold of 0.5 from the mentioned literature. But we would support this further because 0.5 indicates that more than 50% of the data could be predicted with this correlation.

**Figure 7 and 8: The labels on the axes are rather small.**

The label size in Figs. 7 and 8 are increased in the revised manuscript.

**Figure 9: Please use different symbols related to the different colors.**

Figure 9 is changed according to the requirement in the revised manuscript.

**Table 2: What means "NO"?**

"NO" means number. It has been changed to *"#"* for consistency in the revised manuscript.

**Figure 11 and 12: The labels in the figures are much too small and are unreadable.**

We increase the label size and readability of Figs. 11 and 12 in the revised manuscript.

**Figure B1: Please describe the figure in more detail. The particles have the typical shape of NaCl crystals. Please mark the spots in the SEM where the EDX has been recorded. If available an EDX mapping might be shown.**

The particles have a typical sea salt crystal shape as shown in the Fig. C1 in the revised manuscript. Our CCSEM-EDX analysis has covered 0.5 mm by 0.5 mm area on the Al foil substrates to analyze sufficient particle population. Therefore, all particles in Figure C1 have been analyzed by CCSEM-EDX. The figure caption has been extended to describe the image in more detail in the revised manuscript. We have also added Fig. C2 to show the representative EDX-mapping of representative particles in the impinger samples.

**References**

Burkart, J., Gratzl, J., Seifried, T. M., Bieber, P., and Grothe, H.: Isolation of subpollen particles (SPPs) of birch: SPPs are potential carriers of ice nucleating macromolecules, Biogeosciences, 18, 57515765, https://doi.org/10.5194/bg-18-5751-2021, 2021.

Felgitsch, L., Bichler, M., Burkart, J., Fiala, B., Häusler, T., Hitzenberger, R., and Grothe, H.: Heterogeneous Freezing of Liquid Suspensions Including Juices and Extracts from Berries and Leaves from Perennial Plants, Atmosphere-Basel, 10, 37, https://doi.org/10.3390/atmos10010037, 2019.

Fröhlich-Nowoisky, J., Hill, T. C. J., Pummer, B. G., Yordanova, P., Franc, G. D., and Pöschl, U.: Ice nucleation activity in the widespread soil fungus Mortierella alpina, Biogeosciences, 12, 1057 – 1071, https://doi.org/10.5194/bg-12-1057-2015, 2015.

Gute, E. and Abbatt, J. P.: Ice nucleating behavior of different tree pollen in the immersion mode, Atmos. Environ., 231, 117488, https://doi.org/10.1016/j.atmosenv.2020.117488, 2020.

Gute, E., David, R. O., Kanji, Z. A., and Abbatt, J. P.: Ice Nucleation Ability of Tree Pollen Altered by Atmospheric Processing, ACS Earth Space Chem., 4, 12, 2312 – 2319, 2020

**Kozloff, L. M., Turner, M. A., and Arellano, F.: Formation of bacterial membrane ice-nucleating lipoglycoprotein complexes, J. Bacteriol., 173, 6528 – 6536, https://doi.org/10.1128/jb.173.20.65286536.1991, 1991.**

**Pummer, B. G., Bauer, H., Bernardi, J., Bleicher, S., and Grothe, H.: Suspendable macromolecules are responsible for ice nucleation activity of birch and conifer pollen, Atmos. Chem. Phys., 12, 25412550, https://doi.org/10.5194/acp-12-2541-2012, 2012.**

**Seifried, T.M.; Reyzek, F.; Bieber, P.; Grothe, H. Scots Pines (Pinus sylvestris) as Sources of Biological Ice-Nucleating Macromolecules (INMs). Atmosphere 2023, 14, 266. https://doi.org/10.3390/atmos14020266**

References

Koop, T., and Murray, B. J.: A physically constrained classical description of the homogeneous nucleation of ice in water, Chemical Physics., 2016, 145, 1–12, https://doi.org/10.1063/1.4962355.

Hill, T. C. J., DeMott, P. J., Tobo, Y., Fröhlich-Nowoisky, J., Moffett, B. F., Franc, G. D., and Kreidenweis, S. M.: Sources of organic ice nucleating particles in soils, Atmospheric Chemistry and Physics, 16, 7195–7211, https://doi.org/10.5194/acp-16-7195-2016, 2016

Pummer, B. G., Budke, C., Augustin-Bauditz, S., Niedermeier, D., Felgitsch, L., Kampf, C. J., Huber, R. G., Liedl, K. R., Loerting, T., Moschen, T., Schauperl, M., Tollinger, M., Morris, C. E., Wex, H., Grothe, H., Pöschl, U., Koop, T., and Fröhlich-Nowoisky, J.: Ice nucleation by water-soluble macromolecules, Atmospheric Chemistry and Physics, 15, 4077–4091, https://doi.org/10.5194/acp-15-4077-2015, 2015.

Burkart, J., Gratzl, J., Seifried, T. M., Bieber, P., and Grothe, H.: Isolation of subpollen particles (SPPs) of birch: SPPs are potential carriers of ice nucleating macromolecules, Biogeosciences, 18, 57515765, https://doi.org/10.5194/bg-18-5751-2021, 2021.

Seifried, T.M.; Reyzek, F.; Bieber, P.; Grothe, H. Scots Pines (Pinus sylvestris) as Sources of Biological Ice-Nucleating Macromolecules (INMs). Atmosphere 2023, 14, 266. https://doi.org/10.3390/atmos14020266

Beall, C. M., Lucero, D., Hill, T. C., DeMott, P. J., Stokes, M. D., and Prather, K. A.: Best practices for precipitation sample storage for offline studies of ice nucleation in marine and coastal environments, Atmospheric Measurement Techniques, 13, 6473–6486, 2020, https://doi.org/https://doi.org/10.5194/amt-13-6473-2020.

Lacher L, Steinbacher M, Bukowiecki N, Herrmann E, Zipori A, Kanji Z. A.: Impact of Air Mass Conditions and Aerosol Properties on Ice Nucleating Particle Concentrations at the High Altitude Research Station Jungfraujoch. Atmosphere. 2018; 9(9):363. https://doi.org/10.3390/atmos9090363.

Paramonov, M., Drossaart van Dusseldorp, S., Gute, E., Abbatt, J. P. D., Heikkilä, P., Keskinen, J., Chen, X., Luoma, K., Heikkinen, L., Hao, L., Petäjä, T., and Kanji, Z. A.: Condensation/immersion mode ice-nucleating particles in a boreal environment, Atmospheric Chemistry and Physics, 20, 6687–6706, 2020, https://doi.org/10.5194/acp-20-6687-2020.

Rinaldi, M., Hiranuma, N., Santachiara, G., Mazzola, M., Mansour, K., Paglione, M., Rodriguez, C. A., Traversi, R., Becagli, S., Cappelletti, D., et al.: Ice-nucleating particle concentration measurements from Ny-Ålesund during the Arctic spring–summer in 2018, Atmospheric Chemistry and Physics, 21, 14 725–14 748, 2021, https://doi.org/https://doi.org/10.5194/acp-21-14725-2021.

---

## Author Comment (AC2)

**Referee comments 2**

In the study of Li et al. a comprehensive campaign-based study of Ice Nucleating Particles (INPs), physicochemical properties of aerosol particles sampled in Svalbard/Arctic and their source apportionment is presented. It was found that INP composition is different compared to those of aerosol particles and hence aerosol particle composition is not an adequate indicator for INP abundance. Further, heat treatment test of INPs and fluorescence analysis suggest a biological INP origin of an INP subset. The experimental findings in general are sound although not surprisingly new. The mere fact that few atmospheric INP data are available, especially in the Arctic, and the INP sources are not well constrained, justifies the publication of the actual manuscript after careful consideration of the following comments. Generally, the addressed topic fits into the scope of the journal Atmospheric Chemistry and Physics.

We thank referee 2 for the valuable feedback on our manuscript acp-2023-18. In response to the questions and suggestions, please find our answers and revisions listed below. **Referee comments are reproduced in bold** and author responses in normal font; *extracts from the original manuscript are presented in red italic* and *extracts from the revised manuscript in blue italic*.

**General comment:**

**In general, it is not motivated and explained why the two different measurement sites were used to measure aerosol particles and INPs and why the different INP measurement methods DRINCZ and WT-CRAFT operated after different inlet systems. It seems to follow a randomly thrown together measurement set-up rather than a clear concept with a clear vision. I am sure that is not true, but the paper has to be revised in this regard.**

We acknowledge the reviewer's comments and add an additional paragraph in Section 2.2 (see lines 106-111 in the revised manuscript) to explain the different methods and approaches used in the paper: *"To investigate the ambient INP concentrations in immersion-freezing mode, we used different INP sampling and measurement instruments introduced in the following subsections, which provide a large range of sampled particle sizes, time resolutions, freezing temperatures, and hence different INP detection limits (see Table 1). In particular, the droplet-freezing techniques (see Sections 2.2.1 and 2.2.2) have different limits of detection (LOD) due to the different droplet sizes and numbers in the experimental setup."*

In addition, the vision of applying different IN measurement approaches is more clearly addressed in Section 3.3 (see lines 442-444 in the revised manuscript): *"Despite the longer sampling duration (> 3 days) of the WT-CRAFT $N_{INP}$ data, we compare it to the immersion freezing data from DRINCZ because it bridges the temperature gap towards HINC measurements".*

Furthermore, we add motivation sentences to stress the use of WT-CRAFT data and the reasons the INP case studies were selected (see lines 444-448 in the revised manuscript): *"Particularly, one interesting exception highlighted in Fig. 6 (c) (filled symbols labeled as WT-CRAFT_high) displayed both high $N_{INP}$ and heat-resistant INPs for the sample collected from 19/10/2019 to 23/10/2019. To understand the properties of the heat-resistant and high $N_{INP}$ sample in more detail with regard to aerosol properties (e.g., particle sizes), time windows that overlap with the WT-CRAFT_high case that also show a large range of $N_{INP}$ were selected for further investigation (Figures 6 a and b)".*

**In addition, the applied measurement methods raise further questions as the comparison of INP concentration determined from impinger and PM10 sampling is quite different, see comment below. It might help to clearly formulate the concrete scientific questions to be answered first and then explain the approach.**

*"We compare the different approaches to infer the impact of particle size on ice nucleation. INP measurements from different approaches allow us to understand aerosol properties. For example, we could have active INPs from pollen particles above 10 μm, which are not captured by the PM10 measurements, but submicron biogenic macromolecules down to below 100 nm would be. With the impinger, we capture*

*particles larger than 10 μm but no particles smaller than 500 nm. Therefore, different approaches are needed to capture both extreme ends of the size range. Similarly, WT-CRAFT collected particles larger than 0.2 μm but also uses smaller droplet sizes than DRINCZ for freezing experiments and thus can be assessed for ice nucleation temperatures down to -30 ℃ extending the temperature range of the DRINCZ approach (-22 ℃) by 8 ℃. The broader coverage of particle sizes and temperatures measured by the combined methods allows for a better representation of ambient INPs."* We have now clarified this aspect and added the above text in the manuscript on lines 181-188.

**Furthermore, a short paragraph what is known so far about Arctic INPs would be valuable in the introduction.**

For the knowledge about Arctic INPs, it was stressed in the original text regarding their potential sources and origins (see also lines 60-71 in the revised manuscript), but we have reworked the original text to specifically point out where in the Arctic which types of aerosol act as INPs: *"A variety of aerosols of both terrestrial and marine origin in the Arctic can act as INPs in the MPC temperature regime. Mineral dust particles can typically act as INPs at temperatures below approximately -15 ℃ (Kanji et al., 2017; Hoose and Möhler, 2012; Murray et al., 2012). In the Arctic, mineral dust emitted from high latitudes, e.g., from the glacial outwash plains in Svalbard (Tobo et al., 2019), from deserts in Iceland (Sanchez-Marroquin et al., 2020), or dust originating from long-range transport (Vergara-Temprado et al., 2017) are significant terrestrial sources of INPs. In contrast, biological INPs favor heterogeneous ice nucleation at relatively warmer temperatures above approximately -15 ℃ (Murray et al., 2012). Their sources in the Arctic can stem from land, e.g., vegetation (Conen et al., 2016), runoff from watersheds (Tobo et al., 2019) and thawing permafrost (Barry et al., 2023; Creamean et al., 2020) or from the ocean, e.g., sea spray aerosols (SSA) (Irish et al., 2017; DeMott et al., 2016; Wilson et al., 2015), phytoplankton (Ickes et al., 2020; Hartmann et al., 2020; Creamean et al., 2019) and bacterial productivity (Šantl Temkiv et al., 2019). In addition to the INP sources originating from the vicinity of the measurement sites in the local Arctic, the remote effect of INP emissions from mid- to low-latitudes and long-range transport cannot be neglected (Schmale et al., 2021)".*

**Specific comments:**

**P2, L39: Homogeneous ice nucleation has not a fixed onset threshold, it depends on droplet size in addition to temperature. Please adapt wording accordingly.**

We agree with the reviewer. The texts are adjusted from *'In MPCs, where the temperatures are higher than the onset threshold of homogeneous freezing of -38 ℃, primary ice formation...'* to *'In MPCs, where the temperature is higher than the onset of homogeneous freezing at approximately -38 ℃ for cloud droplet relevant sizes, primary ice formation…'* (see lines 38-39 in the revised manuscript). In addition, we also specify the different onset temperatures of homogeneous freezing for our different INP instruments (see caption in Table. 1 in the revised manuscript): *"For the droplet freezing techniques (i.e., DRINCZ and WT-CRAFT), the lower temperature range represents the homogeneous freezing temperature, where pure water is observed to freeze for the corresponding droplet size. In HINC, homogeneous freezing is observed at -38 ℃, which is to be expected given that the droplet sizes are much smaller than the other instruments."*

**Further - more important, since this manuscript is about a comprehensive study of Ice Nucleating Particles, a brief paragraph can be written explaining how an INP is defined and which freezing modes are investigated in the actual study.**

We add statements of definition and relevant freezing mode investigated in the present study: *'Immersion freezing, a heterogeneous freezing process where INPs become immersed in a dilute aqueous solution through the activation of cloud droplets followed by catalyzing freezing from within (Vali et al., 2015), is considered the most important freezing mode in the MPCs (Kanji et al., 2017; Hande and Hoose, 2017; Westbrook and Illingworth, 2013) and will be the focus of this study'* (see lines 41-44 in the revised manuscript).

**P3, L57: To which measure or quantity do refer 'the range of activities as INPs'? The sentence has little meaning in this form.**

We have now changed the sentence *'A variety of aerosols of both terrestrial and marine origin display a range of activities as INPs'* to *'A variety of aerosols of both terrestrial and marine origin in the Arctic can act as INPs in the MPC temperature regime'* (see line 60 in the revised manuscript).

**P3, L59: Dust as a 'significant terrestrial source of INP' could also result from long-range transport as opposed to or in addition to local dust sources. Without further knowledge, this seems unbalanced in the descriptive introduction.**

The role of long-range transport of dust sources is added in the revised manuscript as follows *'Mineral dust emitted from high latitudes, e.g., from the glacial outwash plains in Svalbard (Tobo et al., 2019), from deserts in Iceland (Sanchez-Marroquin et al., 2020), or dust originating from long-range transport (Vergara-Temprado et al., 2017) are significant terrestrial sources of INPs'* (see lines 62-64).

**P5, Fig.2: Which aerosol inlet had been used at the aerosol container? PM10 or TSP inlet? Please specify.**

The aerosol inlet used at the aerosol container is a TSP. It was described in the original text *'The inlet had an upper cut-off threshold of approximately 40 μm (Li et al., 2022) and was heated to a maximum of 40 °C to avoid clogging and frost build-up in the sampling line'* (see lines 89-91 in the revised manuscript).

**P5, section 2.2/ P7, L173-174: A brief introducing paragraph of the complementary INP measurement methods would be helpful to motivate the need and application of the different instruments instead of a very short sentence at the end of the section. In general, statements of which freezing modes are analyzed (only immersion freezing or also deposition nucleation with HINC?), which INP concentration and temperature range can be covered with the applied methods would be needed.**

An introductory paragraph is added in Section 2.2 (lines 106-111 in the revised manuscript) instead of the short sentence at the end of the section (removed): *"To investigate the ambient INP concentrations in immersion-freezing mode, we used different INP sampling and measurement instruments introduced in the following subsections, which provide a large range of sampled particle sizes, time resolutions, freezing temperatures, and hence different INP detection limits (see Table 1). In particular, the droplet-freezing techniques (see Sections 2.2.1 and 2.2.2) have different limits of detection (LOD) due to the different droplet sizes and numbers in the experimental setup."*

In addition, we add Table 1 at the end of Section 2.2 in the revised manuscript, summarizing the features of different INP sampling and measurement techniques.

**P5, L108-109: It is described that the impinger is refilled with clean water continuously in order to compensate the reduced amount of water due to evaporation which would influence the sampling efficiency. This is understandable as well as method to subtract the unwanted INP background from the measurements. However, as impurities might concentrate using the described impinger sampling procedure, it is needed to present the background INP measurements in the appendix of the paper for all applied INP measurement methods for good scientific practice.**

We agree with the reviewer. The background INP spectra were consistently low over the measurement period for the impinger. According to the referee's comment, we added Fig. A1 in the Appendix (Fig. 1 in the current document) to show the assembly of spectra of frozen fraction as a function of temperature for all background and aerosol sample measurements. Figure A1 also contains the same information for the PM10 filters.

[Figure]

Figure 1. Assembly of frozen fraction curves as a function of temperature for aerosol samples and pure water reference experiments conducted with DRINCZ for all measurements from (a) impinger and (b) PM$_{10}$ samples.

**P7, L165: What is meant by 'frost particles'?**
By this we mean ice particles that have grown on the walls of the chamber by vapor diffusion. The original text *'To account for frost particles that can be misidentified as INPs when detaching from the inner surface, we applied a routine of filtered air measurements…'* is changed to *'To account for ice particles emitted from frost build-up, which can be misidentified as INPs when detaching from the inner surface, we applied a routine of filtered air measurements…'* in the revised manuscript for clarification (see lines 173-174).

**P7, L175; P12, Fig. 3 caption, elsewhere in the manuscript: The authors use formulations like 'freezing temperature', 'activation temperature' and similar formulations when they refer to e.g., temperature dependent INP number concentrations. I suggest using the more general formulation of 'temperature' because otherwise the other terms would need a clear definition in the manuscript.**
The original text *"activation temperature"* has been changed to *"temperature"* in the revised manuscript (see e.g., lines 326, 347, 363, Fig. 5 caption and elsewhere).

**P13, L323: What is meant by 'different sensitivities of the instruments'. It needs a specification or an explanation.**
By different sensitivities, we mean different detection thresholds. We now specify this in lines 342-343 of the revised manuscript *"…likely due to different detection threshold of the instruments (i.e., LOD)…"*.

**P14, L335-336: The storage of the INP samples might be one explanation for the observed difference in INP conc., but I could image also other reasons for example different inlet systems, different sampling efficiencies of the different methods etc.**
We agree that in addition to the storage, other factors could contribute to the measurement discrepancies in INPC measured with different systems, including sampling efficiency and particle size threshold as was already addressed in the initial version of the manuscript on lines 340 - 346 (now lines 358-368 in the revised manuscript).

**P14, L338-339: This conclusion is not obvious for the referee as no measurement uncertainty is given in the plot. What would be the implication?**
For the purpose of comparing the range of INPs measured by different methods (i.e., impinger, PM10, and WT-CRAFT) and different treatments (i.e., untreated, heated, and after storage without heating), we plot the median INPC in each category with the vertical extensions representing the 5-95 % percentiles of the measurements. Given that this variation in measurement is larger than the uncertainty it would be more suitable to evaluate the variation. For the degradation due to storage (Fig. 4b), the median values range from being lower, the same or in some cases at the warmest freezing temperatures even higher than those of the fresh samples. As such it is inconclusive that the storage has an effect for the samples measured here.

However, we recommend that this should be verified when long storage is expected before the final analysis of INP samples. Further in Figure 9 where only a subset of samples is shown, the effect of storage is clearer for impinger samples. To better reflect this variability, we have adjusted the statement in the original manuscript on lines 338-339 to *"Similarly, a slight reduction in median INP concentrations was also observed for impinger samples at most of the investigated temperatures when they were stored and reanalyzed in the laboratory (see "after storage" in gray symbols in Fig. 4b). The above reasons, however, would not explain degradation in the PM$_{10}$ samples; as such, we believe that the lower N$_{INP}$ in the PM$_{10}$ samples is indicative of a size dependency since the impinger samples include particles larger than 10 $\mu$m which are excluded in the PM$_{10}$ samples. This conclusion is also supported by the NINP from impinger being systematically higher than those from the PM$_{10}$ samples (see Fig. 3). The NINP of the impinger samples before and after storage largely overlap in freezing temperatures. On conducting a t-test, this difference was insignificant at most investigated temperatures (not shown)"* (see lines 356-363 in the revised manuscript).

**More obvious and striking is that the different methods impinger, PM10 and WT-CRAFT are quite different regarding INP concentration at the same temperature (e.g. -15° C one order of magnitude). Are the INPs collected during the same period? Partly. What are the main sampling differences? Why are these measurements compared? This needs to be more clearly described in the manuscript.**
The INP results shown in this work (Figures 3, 4, and 6) with different approaches (i.e., impinger, PM10, and WT-CRAFT) were measured during the same period in autumn 2019. The differences in INP sampling and measurement techniques are summarized in Table 1 in the revised manuscript, including the sampling site, aerosol size, experimental setup, limit of detection, temperature range, and temporal resolution. The reasons for combining different INP measurements have been already discussed in the response to the first general comment and are addressed throughout the manuscript.

**P14, L343: What is the respective lower limit for PM10? The pore size? And what is the respective upper limit for the impinger? Is anything known of the sampling characteristics in dependence of particle size?**
The PM10 samples were collected on the 0.4 $\mu$m polycarbonate membrane filters. The small-sized particles can be sampled by interception and diffusion, even those below the filter pore size. Therefore, there is no lower size limit because the entire airflow is directed through the filter. This information has been added to the revised manuscript (see footnote in Table 1). The upper size limit for the impinger samples is approximately 20 $\mu$m. The information has been added in Table 1 in the revised manuscript to summarize the characteristics of different INP sampling and measurement techniques.

**P17, L415: What is a 'peak case'? Do the authors refer to a higher INP number conc. or fraction? Please specify.**
It refers to the case with a higher INP number concentration, which is specified in line 434 in the revised manuscript.

**P18, L418 and elsewhere in the manuscript: It would be helpful to motivate the investigations or analysis which have been done and are described in each section.**
Motivational sentences have been added at the beginning of each subsection in the Results section (see lines 432 and 452 in the revised manuscript).

**P20, L448-450: I cannot agree with this statement in its current form that the heat-labile INPs would be overshadowed by heat stable INPs. It is more likely, in my opinion that no heat sensible INPs are present in that sample. Maybe the heat sensible INP are not present in the air during that period, or a longer sampling period would have been required to collect even very rare INPs. In principle, it can be possible that very frequent INPs overlap with less frequent INPs and it would not be possible to discriminate between both populations except for the case that the less frequent are more active in nucleating ice, i.e., initiate freezing at higher temperature, because the most efficient (regarding temperature) INP determine the freezing of a droplet. Please rephrase this statement accordingly or clarify your statement.**
We appreciate the reviewer's comment, however, the offered explanation would not explain our observations as the sample that showed an absence of heat-labile INPs was the one with the longest sampling time. The current statement is based on the comparison between relatively high-INP and heat-resistant WT-

CRAFT sample (4-day resolution, filled circles in Fig. 4c) and the parallel PM10 samples (8-h resolution, i.e., PM10_high, PM10_moderate, and PM10_low included). The longer sampling time for WT-CRAFT would imply that heat-labile INPs should have been sampled over the four days because the PM10 and impinger samples were collected over shorter sampling times during overlapping period at the location of the container and the GVB (1 km away) both showed signatures of heat-labile INPs. So the possibility that the total of heat-labile INPs is not sufficient during the 4-day measurement would not explain this observation. The reason that the reviewer pointed out (see underlined text above), is an eloquent way of explaining the potential shadowing of the heat-labile INPs by the heat-resistant ones. I.e. because the lower time resolution samples collect aerosol over 4 days, it is possible that in one droplet heat-labile and heat resistant INPs co-exist, and thus the most efficient INP determines the freezing of the droplet. Thus even if the heat labile INP is degraded in this sample, the heat-stable one still nucleates ice, masking the effect of the heat-labile sample in the lower time resolution WT-CRAFT data.

As such we agree with the referee that the most efficient INPs (i.e., PM10_high) determine the droplet freezing, leading to an overall high INPC in a coarser-resolution sample (i.e., the abovementioned WT-CRAFT sample). Therefore, we adjusted the relevant statement in the revised manuscript (lines 481-489): *'PM10_low was an exception, showing heat-resistant INP composition, consistent with the heating results of the corresponding WT-CRAFT sample from the GVB station (filled circles in Fig. 4c), where the 4-day WT-CRAFT sample possessed relatively high $N_{INP}$ with heat-resistance. On the other hand, the parallel PM10 cases with much finer temporal resolution (i.e., 8-h PM10_high, PM10_moderate and PM10_low samples) covered a wide range (i.e., over 2 orders of magnitude) of $N_{INP}$ with different sensitivities to heat treatment (see the second row in Fig. 9). Since $N_{INP}$ is determined by the most efficient INPs in the droplet, samples that contain heat-labile INPs and heat-resistant INPs could still freeze as efficiently after heating, thus masking the effect of heating. Sampling with higher time resolution reduces the probability of including INPs of different properties within the same droplet, thus motivating finer temporal resolution of INP measurements in field studies that desire the characterization of INP properties.'*

**P21, Fig. 9: Measurement uncertainties are missing in the plot. Please add those.**
The 95 % confidence intervals are added, and the updated figure is shown below (Fig. 9 in the revised manuscript).

[Figure]

Figure 2. Selected INP spectra for untreated, heated, and storage conditions labeled high, moderate, and low $N_{INP}$ for case studies for impinger and $PM_{10}$ samples. The vertical extensions represent the 95 % confidence intervals of the experiments. Note that the time stamps for the same $N_{INP}$ labels regarding the impinger and $PM_{10}$ samples are different. High INP cases: 11:05-12:05 UTC 21/10/2019 and 00:00-08:00 UTC 19/10/2019, moderate INP cases: 06:34-07:34 UTC 24/10/2019 and 16:00-24:00 UTC 21/10/2019, and low INP cases: 15:40-16:40 UTC 25/10/2019 and 00:00-08:00 UTC 23/10/2019 for impinger and $PM_{10}$ samples, respectively.

**To classify the PM10 and the impinger method regarding high, moderate and low INP load for comparison seems too reasonable at first. However, it distracts from the question why different methods lead to different INP concentrations and which one are more representative for atmospheric INP concentration?**
We address this in Section 4 (see lines 537-544 in the revised manuscript) as follows: *"In this work, despite different INP concentrations being observed by the applied INP techniques (i.e., HINC, $PM_{10}$, impinger, and WT-CRAFT), all methods are representative in the context of different properties of collected aerosols. A range of aerosol particle sizes can act as INPs so that different measurements are needed to cover the full-size range of aerosols smaller than 20 µm. This is true for particle size distribution measurements as well. $PM_{10}$ filter and WT-CRAFT collect aerosol particles below 10 µm, and the impinger samples particles between 0.5 and 20 µm. A broader range of particle sizes measured by the combined methods allows for a better representation of ambient INPs. In addition, INP measurements from different approaches allow us to determine INP properties from different species. For instance, if one focuses exclusively on mineral dust, a method is needed that captures the coarse-mode aerosol particles."*

**Which one can be used as model input data? Or is this effect an artifact from different measurement sites? But then the question arise why the measurement set-up was used as it was, i.e., measurement containers with different inlets at different locations with different instrumentation. What is the scientific question underlying this approach?**
Measurements from all methods can be used as model input data depending on the aerosol properties and research objectives as discussed in the previous answer. In general, they could be applied together in models as the natural variation in INPC is also quite high (i.e., the lower end of the INPC range from the PM10 samples and the highest from the impinger samples). Besides, from Fig. 3 one can tell that *"...the distribution of WT-CRAFT INP data overlaps largely with $PM_{10}$ and impinger data, and the data are approximately log-linearly*

*extrapolatable, although some variation is observed due to the limited sample number and coarse temporal resolution"* (added in lines 328-331 in the revised manuscript). Additionally, in Fig. 6, *"The N$_{INP}$ from the impinger measurements is higher than that measured from the WT-CRAFT. This could be either due to the different locations of the samples or the larger drop sizes in DRINCZ. The total volume of air sampled for the two samples is similar. The more likely explanation is that the particle size range measured by the impinger is much larger. As such, the sampled aerosol size ranges should be considered when evaluating such comparisons."* (Added in lines 438-442 in the revised manuscript).

**P22, Fig. 10: Can we conclude from the bulk aerosol chemical composition that different aerosols are analyzed at the 2 different measurement sites?**
Generally, the air masses should be identical at the two measurement sites (i.e., the aerosol container and GVB station) given they are 1 km apart, despite their different wind patterns likely caused by local topography (see Fig. 1b in the revised manuscript). Initially one could be tempted to conclude as the reviewer suggests that different aerosols are analysied at the two measurement sites because the high, moderate, and low N$_{INP}$ cases from the impinger and PM10 are not occurring at identical times. However, given the different size ranges targeted by the impinger and PM10 samples, we would need to be careful with such a conclusion because we know for sure different size ranges are sampled. We have now added the following statement to the revised manuscript on lines 495-497 *"Note that the air masses cannot be compared directly from the compositions of the high, moderate, and low N$_{INP}$ cases from the impinger and PM$_{10}$ samples in (Fig.10) because they were not taken at the same time."*

**P25, L518: Please add a statement if long-term observations would be needed to obtain a better statistical base for investigation also regarding evaluation approached which are based on correlation methods.**
Thanks. The original sentence is changed to *'Further studies with long-term observations are needed to elucidate the annual sources of INPs in the Arctic on a better statistical basis.'* (see lines 565-566 in the revised manuscript).

**Technical correction:**

**P2, L38: Change 'temperatures are higher' to 'temperature is higher'.**
Thanks. Change was made according to the comment (see line 38 in the revised manuscript).

**P3, L62: 'sediments from runoff' to what area do you refer 'runoff from the watershed'?**
Thanks. Change was made according to the comment (see line 66 in the revised manuscript).

**P6, L139: Change 'estimate' to 'measure'.**
Thanks. Change was made according to the comment (see line 147 in the revised manuscript).

**P7, L163: Remove 'Fig. 2 of'.**
Thanks. Relevant text is removed accordingly (see line 171 in the revised manuscript).

**P20, L451: Remove 'despite the increased labor intensity'.**
Thanks. Relevant text is removed accordingly.

**P25, L510-511: What is meant by 'high wind speed of marine origin'? Please reformulate!**
Thanks. Former text has been changed to *'high wind speed originating from the ocean'* (see lines 557-558 in the revised manuscript).